# Solving Linear Inverse Problems Provably via Posterior Sampling with Latent Diffusion Models

Litu Rout    Negin Raoof    Giannis Daras

Constantine Caramanis    Alexandros G. Dimakis    Sanjay Shakkottai

The University of Texas at Austin*

## Abstract

We present the first framework to solve linear inverse problems leveraging pre-trained *latent* diffusion models. Previously proposed algorithms (such as DPS and DDRM) only apply to *pixel-space* diffusion models. We theoretically analyze our algorithm showing provable sample recovery in a linear model setting. The algorithmic insight obtained from our analysis extends to more general settings often considered in practice. Experimentally, we outperform previously proposed posterior sampling algorithms in a wide variety of problems including random inpainting, block inpainting, denoising, deblurring, destriping, and super-resolution.

## 1 Introduction

We study the use of pre-trained latent diffusion models to solve linear inverse problems such as denoising, inpainting, compressed sensing and super-resolution. There are two classes of approaches for inverse problems: supervised methods where a restoration model is trained to solve the task at hand [37, 39, 56, 31], and unsupervised methods that use the prior learned by a generative model to guide the restoration process [52, 40, 5, 33, 11, 26]; see also the survey of Ongie et al. [36] and references therein.

The second family of unsupervised methods has gained popularity because: (i) general-domain foundation generative models have become widely available, (ii) unsupervised methods do not require any training to solve inverse problems and leverage the massive data and compute investment of pre-trained models and (iii) generative models *sample* from the posterior-distribution, mitigating certain pitfalls of likelihood-maximization methods such as bias in the reconstructions [35, 24] and regression to the mean [23, 22].

Diffusion models have emerged as a powerful new approach to generative modeling [47, 48, 49, 20, 29, 18, 54]. This family of generative models works by first corrupting the data distribution $p_0(\boldsymbol{x}_0)$ using an Itô Stochastic Differential Equation (SDE), $\mathrm{d}\boldsymbol{x} = \boldsymbol{f}(\boldsymbol{x}, t)\mathrm{d}t + g(t)\mathrm{d}\boldsymbol{w}$, and then by learning the score-function, $\nabla_{\boldsymbol{x}_t} \log p_t(\boldsymbol{x}_t)$, at all levels $t$, using Denoising Score Matching (DSM) [21, 53]. The seminal result of Anderson [1] shows that we can reverse the corruption process, i.e., start with noise and then sample from the data distribution, by running another Itô SDE. The SDE that corrupts the data is often termed as Forward SDE and its reverse Reverse SDE [49]. The latter depends on the score-function $\nabla_{\boldsymbol{x}_t} \log p_t(\boldsymbol{x}_t)$ that we learn through DSM. In [8, 9], the authors provided a non-asymptotic analysis for the sampling of diffusion models when the score-function is only learned approximately.

The success of diffusion models sparked the interest to investigate how we can use them to solve inverse problems. Song et al. [49] showed that given measurements $\boldsymbol{y} = \mathcal{A}\boldsymbol{x}_0 + \sigma_y\boldsymbol{n}$, we can

---

*Email:{litu.rout,neginmr,giannisdaras,constantine,sanjay.shakkottai}utexas.edu, dimakis@austin.utexas.edu

37th Conference on Neural Information Processing Systems (NeurIPS 2023).

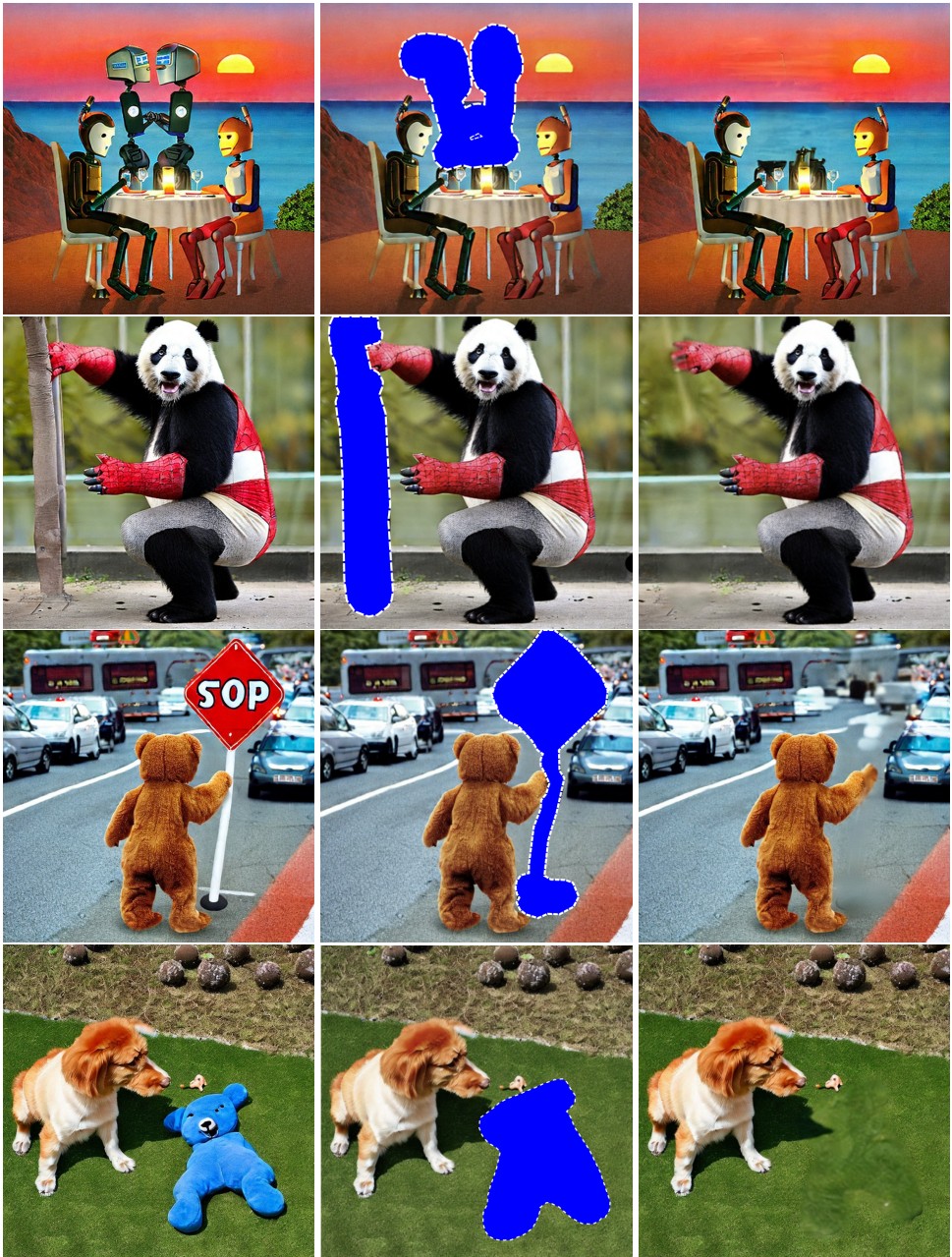

Figure 1: Overall pipeline of our proposed framework from left to right. Given an image (**left**) and a user defined mask (**center**), our algorithm inpaints the masked region (**right**). The known part of the images are unaltered (see Appendix C for web demo and image sources).

provably sample from the distribution $p_0(\boldsymbol{x}_0|\boldsymbol{y})$ by running a modified Reverse SDE that depends on the unconditional score $\nabla_{\boldsymbol{x}_t} \log p_t(\boldsymbol{x}_t)$ and the term $\nabla_{\boldsymbol{x}_t} \log p(\boldsymbol{y}|\boldsymbol{x}_t)$. The latter term captures how much the current iterate explains the measurements and it is intractable even for linear inverse problems without assumptions on the distribution $p_0(x_0)$ [11, 14]. To deal with the intractability of the problem, a series of approximation algorithms have been developed [22, 11, 2, 13, 26, 10, 6, 46, 12, 27] for solving (linear and non-linear) inverse problems with diffusion models. These algorithms use pre-trained diffusion models as flexible priors for the data distribution to effectively solve problems such as inpainting, deblurring, super-resolution among others.

Recently, diffusion models have been generalized to learn to invert non-Markovian and non-linear corruption processes [16, 15, 3]. One instance of this generalization is the family of Latent Diffusion Models (LDMs) [41]. LDMs project the data into some latent space, $\boldsymbol{z}_0 = \mathcal{E}(\boldsymbol{x}_0)$, perform the

diffusion in the latent space and use a decoder, $\mathcal{D}(\boldsymbol{z}_0)$, to move back to the pixel space. LDMs power state-of-the-art foundation models such as Stable Diffusion [41] and have enabled a wide-range of applications across many data modalities including images [41], video [4], audio [30] and medical domain distributions (e.g., for MRI and proteins) [38, 51]. Unfortunately, none of the existing algorithms for solving inverse problems works with Latent Diffusion Models. Hence, to use a foundation model, such as Stable Diffusion, for some inverse problem, one needs to perform finetuning for each task of interest.

In this paper, we present the first framework to solve general inverse problems with pre-trained *latent* diffusion models. Our main idea is to extend DPS by adding an extra gradient update step to guide the diffusion process to sample latents for which the decoding-encoding map is not lossy. By harnessing the power of available foundation models, we are able to outperform previous approaches without finetuning across a wide range of problems (see Figure 1 and 2).

**Our contributions are as follows:**

(i) We show how to use Latent Diffusion Models models (such as Stable Diffusion) to solve linear inverse problem when the degradation operator is known.

(ii) We theoretically analyze our algorithm and show provable sample recovery in a linear model setting with two-step diffusion processes.

(iii) We achieve a new state-of-the-art for solving inverse problems with latent diffusion models, outperforming previous approaches for inpainting, block inpainting, denoising, deblurring, destriping, and super-resolution.[2]

## 2 Background and Method

**Notation:** Bold lower-case $\boldsymbol{x}$, bold upper-case $\boldsymbol{X}$, and normal lower case $x$ denote a vector, a matrix, and a scalar variable, respectively. We denote by $\odot$ element-wise multiplication. $\boldsymbol{D}(\boldsymbol{x})$ represents a diagonal matrix with entries $\boldsymbol{x}$. We use $\mathcal{E}(.)$ for the encoder and $\mathcal{D}(.)$ for the decoder. $\mathcal{E}\sharp p$ is a pushforward measure of $p$, i.e., for every $\boldsymbol{x} \in p$, the sample $\mathcal{E}(\boldsymbol{x})$ is a sample from $\mathcal{E}\sharp p$. We use arrows in Section 3 to distinguish random variables of the forward ($\rightarrow$) and the reverse process ($\leftarrow$).

The standard diffusion modeling framework involves training a network, $\boldsymbol{s}_\theta(\boldsymbol{x}_t, t)$, to learn the score-function, $\nabla_{\boldsymbol{x}_t} \log p_t(\boldsymbol{x}_t)$, at all levels $t$, of a stochastic process described by an Itô SDE:

$$\mathrm{d}\boldsymbol{x} = \boldsymbol{f}(\boldsymbol{x}, t)\mathrm{d}t + g(t)\mathrm{d}\boldsymbol{w}, \tag{1}$$

where $\boldsymbol{w}$ is the standard Wiener process. To generate samples from the trained model, one can run the (unconditional) Reverse SDE, where the score-function is approximated by the trained neural network. Given measurements $\boldsymbol{y} = \mathcal{A}x_0 + \sigma_y \boldsymbol{n}$, one can sample from the distribution $p_0(\boldsymbol{x}_0|\boldsymbol{y})$ by running the conditional Reverse SDE given by:

$$\mathrm{d}\boldsymbol{x} = \left(\boldsymbol{f}(\boldsymbol{x}, t) - g^2(t)\left(\nabla_{\boldsymbol{x}_t} \log p_t(\boldsymbol{x}_t) + \nabla_{\boldsymbol{x}_t} \log p(\boldsymbol{y}|\boldsymbol{x}_t)\right)\right)\mathrm{d}t + g(t)\mathrm{d}\boldsymbol{w}. \tag{2}$$

As mentioned, $\nabla_{\boldsymbol{x}_t} \log p(\boldsymbol{y}|\boldsymbol{x}_t)$ is intractable for general inverse problems. One of the most effective approximation methods is the DPS algorithm proposed by Chung et al. [11]. DPS assumes that:

$$p(\boldsymbol{y}|\boldsymbol{x}_t) \approx p\left(\boldsymbol{y}|\hat{\boldsymbol{x}}_0 \coloneqq \mathbb{E}[\boldsymbol{x}_0|\boldsymbol{x}_t]\right) = \mathcal{N}(\boldsymbol{y}; \mu = \mathcal{A}\mathbb{E}[\boldsymbol{x}_0|\boldsymbol{x}_t], \Sigma = \sigma_y^2 I). \tag{3}$$

Essentially, DPS substitutes the unknown clean image $\boldsymbol{x}_0$ with its conditional expectation given the noisy input, $\mathbb{E}[\boldsymbol{x}_0|\boldsymbol{x}_t]$. Under this approximation, the term $p(\boldsymbol{y}|\boldsymbol{x}_t)$ becomes tractable.

The theoretical properties of the DPS algorithm are not well understood. In this paper, we analyze DPS in a linear model setting where the data distribution lives in a low-dimensional subspace, and show that DPS actually samples from $p(\boldsymbol{x}_0|\boldsymbol{y})$ (Section A.1). Then, we provide an *algorithm* (Section 2.1) and its *analysis* to sample from $p(\boldsymbol{x}_0|\boldsymbol{y})$ using latent diffusion models (Section 3.2). Importantly, our analysis suggests that our algorithm enjoys the same theoretical guarantees while avoiding the curse of ambient dimension observed in pixel-space diffusion models including DPS. Using experiments (Section 4), we show that our algorithm allows us to use powerful foundation models and solve linear inverse problems, outperforming previous unsupervised approaches without the need for finetuning.

---

[2]The source code is available at: `https://github.com/LituRout/PSLD` and a web application for image inpainting is available at: `https://huggingface.co/spaces/PSLD/PSLD`.

## 2.1 Method

In Latent Diffusion Models, the diffusion occurs in the latent space. Specifically, we train a model $s_\theta(z_t, t)$ to predict the score $\nabla_{z_t} \log p_t(z_t)$, of a diffusion process:

$$d z = f(z, t) dt + g(t) dw, \tag{4}$$

where $z_0 = \mathcal{E}(x_0)$ for some encoder function $\mathcal{E}(\cdot) : \mathbb{R}^d \to \mathbb{R}^k$. During sampling, we start with $z_T$, we run the Reverse Diffusion Process and then we obtain a clean image by passing $z_0 \sim p_0(z_0|z_T)$ through a decoder $\mathcal{D} : \mathbb{R}^k \to \mathbb{R}^d$.

Although Latent Diffusion Models underlie some of the most powerful foundation models for image generation, existing algorithms for solving inverse problems with diffusion models do not apply for LDMs. The most natural extension of the DPS idea would be to approximate $p(y|z_t)$ with:

$$p(y|z_t) \approx p(y|x_0 = \mathcal{D}(\mathbb{E}[z_0|z_t])), \tag{5}$$

i.e., to approximate the unknown clean image $x_0$ with the decoded version of the conditional expectation of the clean latent $z_0$ given the noisy latent $z_t$. However, as we show experimentally in Section 4, this idea does not work. The failure of the "vanilla" extension of the DPS algorithm for latent diffusion models should not come as a surprise. The fundamental reason is that the encoder is a many-to-one mapping. Simply put, there are many latents $z_0$ that correspond to encoded versions of images that explain the measurements. Taking the gradient of the density given by (5) could be pulling $z_t$ towards any of these latents $z_0$, potentially in different directions. On the other hand, the score-function is pulling $z_t$ towards a specific $z_0$ that corresponds to the best denoised version of $z_t$.

To address this problem, we propose an extra term that penalizes latents that are not fixed-points of the composition of the decoder-function with the encoder-function. Specifically, we approximate the intractable $\nabla \log p(y|z_t)$ with:

$$\nabla_{z_t} \log p(y|z_t) = \underbrace{\nabla_{z_t} \log p(y|\hat{x}_0 = \mathcal{D}(\mathbb{E}[z_0|z_t]))}_{\text{DPS vanilla extension}} + \gamma_t \underbrace{\nabla_{z_t} ||\mathbb{E}[z_0|z_t] - \mathcal{E}(\mathcal{D}(\mathbb{E}[z_0|z_t]))||^2}_{\text{"goodness" of } z_0}. \tag{6}$$

We refer to this approximation as Goodness Modified Latent DPS (GML-DPS). Intuitively, we guide the diffusion process towards latents such that: i) they explain the measurements when passed through the decoder, and ii) they are fixed points of the decoder-encoder composition. The latter is useful to make sure that the generated sample remains on the manifold of real data. However, it does not penalize the reverse SDE for generating other latents $z_0$ as long as $\mathcal{D}(z_0)$ lies on the manifold of natural images. Even in the linear case (see Section 3), this can lead to inconsistency at the boundary of the mask in the pixel space. The linear theory in Section 3 suggests that we can circumvent this problem by introducing the following gluing objective. In words, the gluing objective penalizes decoded images having a discontinuity at the boundary of the mask.

$$\nabla_{z_t} \log p(y|z_t) = \underbrace{\nabla_{z_t} \log p(y|x_0 = \mathcal{D}(\mathbb{E}[z_0|z_t]))}_{\text{DPS vanilla extension}}$$
$$+ \gamma_t \underbrace{\nabla_{z_t} ||\mathbb{E}[z_0|z_t] - \mathcal{E}(\mathcal{A}^T y + (I - \mathcal{A}^T \mathcal{A})\mathcal{D}(\mathbb{E}[z_0|z_t]))||^2}_{\text{"gluing" of } z_0}. \tag{7}$$

The gluing objective is critical for our algorithm as it ensures that the denoising update, measurement-matching update, and the gluing update point to the same optima in the latent space. We refer to this approximation (7) as Posterior Sampling with Latent Diffusion (PSLD). In the next Section 3, we provide an analysis of these gradient updates, along with the associated algorithms.

**Remark 2.1.** Consider the optimization problem of projecting onto the measurements:

$$\min_{x_0} ||\hat{x}_0 - x_0||_2^2$$
$$\text{subject to } \mathcal{A}x_0 = y,$$

In the linear setting, the optimal solution is given by $x_0^* = \mathcal{A}^T(\mathcal{A}\mathcal{A}^T)^{-1}y + (\hat{x}_0 - \mathcal{A}^T(\mathcal{A}\mathcal{A}^T)^{-1}(\mathcal{A}\hat{x}_0))$. Now further suppose that the measurement rows are orthogonal, i.e. $\mathcal{A}\mathcal{A}^T = I_l$. This condition holds for some natural linear inverse problems like inpainting. Suppose that we want to update the latent vector $z_t$ such that $\mathbb{E}[z_0|z_t] = \mathcal{E}(x_0^*)$; this ensures that the gradients

| **Algorithm 1:** DPS | **Algorithm 2:** PSLD |
|---|---|
| **Input:** $T, \boldsymbol{y}, \zeta_{i=1}^T, \{\tilde{\sigma}_i\}_{i=1}^T, \boldsymbol{s}_\theta$ | **Input:** $T, \boldsymbol{y}, \{\eta_i\}_{i=1}^T, \{\gamma_i\}_{i=1}^T, \{\tilde{\sigma}_i\}_{i=1}^T, \mathcal{E}, \mathcal{D}, \mathcal{A}, \boldsymbol{s}_\theta$ |

$$
\begin{aligned}
&\textbf{1 } \boldsymbol{x}_T \sim \mathcal{N}(\boldsymbol{0}, \boldsymbol{I}) \\
&\textbf{2 for } i = T-1 \textbf{ to } 0 \textbf{ do} \\
&\textbf{3 } \quad \hat{\boldsymbol{s}} \leftarrow \boldsymbol{s}_\theta(\boldsymbol{x}_i, i) \\
&\textbf{4 } \quad \hat{\boldsymbol{x}}_0 \leftarrow \frac{1}{\sqrt{\bar{\alpha}_i}}(\boldsymbol{x}_i + (1-\bar{\alpha}_i)\hat{\boldsymbol{s}}) \\
&\textbf{5 } \quad \boldsymbol{z} \sim \mathcal{N}(\boldsymbol{0}, \boldsymbol{I}) \\
&\textbf{6 } \quad \boldsymbol{x}'_{i-1} \leftarrow \\
&\qquad \frac{\sqrt{\bar{\alpha}_i}(1-\bar{\alpha}_{i-1})}{1-\bar{\alpha}_i}\boldsymbol{x}_i + \frac{\sqrt{\bar{\alpha}_{i-1}}\beta_i}{1-\bar{\alpha}_i}\hat{\boldsymbol{x}}_0 + \tilde{\sigma}_i \boldsymbol{z} \\
&\textbf{7 } \quad \boldsymbol{x}_{i-1} \leftarrow \boldsymbol{x}'_{i-1} - \zeta_i \nabla_{\boldsymbol{x}_i} \|\boldsymbol{y} - \mathcal{A}(\hat{\boldsymbol{x}}_0)\|_2^2 \\
&\textbf{8 end} \\
&\textbf{9 return } \hat{\boldsymbol{x}}_0
\end{aligned}
$$

$$
\begin{aligned}
&\textbf{1 } \boldsymbol{z}_T \sim \mathcal{N}(\boldsymbol{0}, \boldsymbol{I}) \\
&\textbf{2 for } i = T-1 \textbf{ to } 0 \textbf{ do} \\
&\textbf{3 } \quad \hat{\boldsymbol{s}} \leftarrow \boldsymbol{s}_\theta(\boldsymbol{z}_i, i) \\
&\textbf{4 } \quad \hat{\boldsymbol{z}}_0 \leftarrow \frac{1}{\sqrt{\bar{\alpha}_i}}(\boldsymbol{z}_i + (1-\bar{\alpha}_i)\hat{\boldsymbol{s}}) \\
&\textbf{5 } \quad \boldsymbol{\epsilon} \sim \mathcal{N}(\boldsymbol{0}, \boldsymbol{I}) \\
&\textbf{6 } \quad \boldsymbol{z}'_{i-1} \leftarrow \frac{\sqrt{\bar{\alpha}_i}(1-\bar{\alpha}_{i-1})}{1-\bar{\alpha}_i}\boldsymbol{z}_i + \frac{\sqrt{\bar{\alpha}_{i-1}}\beta_i}{1-\bar{\alpha}_i}\hat{\boldsymbol{z}}_0 + \tilde{\sigma}_i \boldsymbol{\epsilon} \\
&\textbf{7 } \quad \boldsymbol{z}''_{i-1} \leftarrow \boldsymbol{z}'_{i-1} - \eta_i \nabla_{\boldsymbol{z}_i} \|\boldsymbol{y} - \mathcal{A}(\mathcal{D}(\hat{\boldsymbol{z}}_0))\|_2^2 \\
&\textbf{8 } \quad \boldsymbol{z}_{i-1} \leftarrow \\
&\qquad \boldsymbol{z}''_{i-1} - \gamma_i \nabla_{\boldsymbol{z}_i} \|\hat{\boldsymbol{z}}_0 - \mathcal{E}(\mathcal{A}^T\boldsymbol{y} + (\boldsymbol{I} - \mathcal{A}^T\mathcal{A})\mathcal{D}(\hat{\boldsymbol{z}}_0))\|_2^2 \\
&\textbf{9 end} \\
&\textbf{10 return } \mathcal{D}(\hat{\boldsymbol{z}}_0)
\end{aligned}
$$

resulting from the two terms in (7) both point to the same optima in the latent space. Equivalently, we want to solve the following minimization problem: $\min_{\boldsymbol{z}_t} \|\mathbb{E}[\boldsymbol{z}_0|\boldsymbol{z}_t] - \mathcal{E}(x_0^*)\|_2^2$. Substituting $\mathcal{E}(x_0^*) = \mathcal{E}(\mathcal{A}^T\boldsymbol{y} + (\hat{\boldsymbol{x}}_0 - \mathcal{A}^T\mathcal{A}\hat{\boldsymbol{x}}_0)) = \mathcal{E}(\mathcal{A}^T\boldsymbol{y} + (\boldsymbol{I} - \mathcal{A}^T\mathcal{A})\hat{\boldsymbol{x}}_0)$, and $\hat{\boldsymbol{x}}_0 = \mathcal{D}(\mathbb{E}[\boldsymbol{z}_0|\boldsymbol{z}_t])$, we can thus interpret the gluing objective in (7) as a one step of gradient descent of this loss $\|\mathbb{E}[\boldsymbol{z}_0|\boldsymbol{z}_t] - \mathcal{E}(x_0^*)\|_2^2$ with respect to $\boldsymbol{z}_t$. Note that, if there was no latent space, our gluing would be equivalent to a projection on the measurements, but now because of the encoder and decoder, it is not.

## 3 Theoretical Results

As discussed in Section 2, diffusion models consist of two stochastic processes: the forward and reverse processes, each governed by Itô SDEs. For implementation purposes, these SDEs are discretized over a finite number of (time) steps, and the diffusion takes place using a transition kernel. The forward process starts from $\overrightarrow{\boldsymbol{x}_0} \sim p(\overrightarrow{\boldsymbol{x}_0})$ and gradually adds noise, i.e., $\overrightarrow{\boldsymbol{x}}_{t+1} = \sqrt{1-\beta_t}\overrightarrow{\boldsymbol{x}}_t + \sqrt{\beta_t}\boldsymbol{\epsilon}$ where $\beta_t \in [0,1]$ and $\beta_t \geq \beta_{t-1}$ for $t = 0, \ldots, T-1$ . The reverse process is initialized with $\overleftarrow{\boldsymbol{x}}_T \sim \mathcal{N}(\boldsymbol{0}, \boldsymbol{I}_d)$ and generates $\overleftarrow{\boldsymbol{x}}_{t-1} = \mu_\theta(\overleftarrow{\boldsymbol{x}}_t, t) + \sqrt{\beta_t}\boldsymbol{\epsilon}$. In the last step, $\mu_\theta(\overleftarrow{\boldsymbol{x}}_1, 1)$ is displayed without the noise.

In this section, we consider the diffusion discretized to two steps ($\{\overrightarrow{\boldsymbol{x}_0}, \overrightarrow{\boldsymbol{x}_1}\}$), and a Gaussian transition kernel that arises from the Ornstein-Uhlenbeck (OU) process. We choose this setup because it captures essential components of complex diffusion processes without raising unnecessary complications in the analysis. We provide a principled analysis of **Algorithm 1** and **Algorithm 2** in a linear model setting with this two-step diffusion process under assumptions that guarantee exact reconstruction is possible in principle. A main result of our work is to prove that in this setting we can solve inverse problems perfectly. As we show, this requires some novel algorithmic ideas that are suggested by our theory. In Section 4, we then show that these algorithmic ideas are much more general, and apply to large-scale real-world applications of diffusion models that use multiple steps ($\{\overrightarrow{\boldsymbol{x}_0}, \overrightarrow{\boldsymbol{x}_1}, \cdots, \overrightarrow{\boldsymbol{x}_T}\}$, where $T = 1000$), and moreover do not satisfy the recoverability assumptions. We provide post-processing details of **Algorithm 2** in Appendix C.1. All proofs are given in Appendix B.

### 3.1 Problem Setup

The goal is to show that posterior sampling algorithms (such as DPS) can provably solve inverse problems in a perfectly recoverable setting. To show exact recovery, we analyze two-step diffusion processes in a linear model setting similar to [42, 7], where the images ($\overrightarrow{\boldsymbol{x}_0} \in \mathbb{R}^d$) reside in a linear subspace of the form $\overrightarrow{\boldsymbol{x}_0} = \mathcal{S}\overrightarrow{\boldsymbol{w}_0}, \mathcal{S} \in \mathbb{R}^{d \times l}, \overrightarrow{\boldsymbol{w}_0} \in \mathbb{R}^l$, and $\sigma_y = 0$. Here, $\mathcal{S}$ is a tall thin matrix with $rank(\mathcal{S}) = l \leq d$ that lifts any latent vector $\overrightarrow{\boldsymbol{w}_0} \sim \mathcal{N}(\boldsymbol{0}, \boldsymbol{I}_l)$ to the image space with ambient dimension $d$. Given the measurements $\boldsymbol{y} = \mathcal{A}\overrightarrow{\boldsymbol{x}_0} + \sigma_y \boldsymbol{n}, \mathcal{A} \in \mathbb{R}^{l \times d}, \boldsymbol{n} \in \mathbb{R}^l$, the goal is to sample from $p_0(\overrightarrow{\boldsymbol{x}_0}|\boldsymbol{y})$ using a pre-trained latent diffusion model. In the inpainting task, the measurement operator $\mathcal{A}$ is such that $\mathcal{A}^T\mathcal{A}$ is a diagonal matrix $\boldsymbol{D}(\boldsymbol{m})$, where $\boldsymbol{m}$ is the masking vector with elements set to 1 where data is observed and 0 where data is masked (see Appendix B for further details). Recall that in latent diffusion models, the diffusion takes place in the latent space of a pre-trained Variational Autoencoder (VAE). Following the common practice [41], we consider a setting where the latent vector of the VAE is $k$-dimensional and the latent distribution is a standard Gaussian $\mathcal{N}(\boldsymbol{0}, \boldsymbol{I}_k)$. Our analysis shows that the proposed **Algorithm 2** provably solves inverse problems under the following assumptions.

**Assumption 3.1.** The columns of the data generating model $\mathcal{S}$ are orthonormal, i.e., $\mathcal{S}^T \mathcal{S} = \boldsymbol{I}_l$.

**Assumption 3.2.** The measurement operator $\mathcal{A}$ satisfies $(\mathcal{A}\mathcal{S})^T(\mathcal{A}\mathcal{S}) \succ \boldsymbol{0}$.

These assumptions have previously appeared, e.g., [42]. While **Assumption 3.1** is mild and can be relaxed at the expense of (standard) mathematical complications, **Assumption 3.2** indicates that $(\mathcal{A}\mathcal{S})^T(\mathcal{A}\mathcal{S})$ is a positive definite matrix. The latter ensures that there is enough energy left in the measurements for perfect reconstruction. More precisely, any subset of $l$ coordinates exactly determines the remaining $(d - l)$ coordinates of $\overrightarrow{\boldsymbol{x}_0}$. The underlying assumption is that there *exists* a solution and it is *unique* [42]. Thus, the theoretical question becomes how close the recovered sample is to this groundtruth sample from the true posterior. Alternatively, one may consider other types of posteriors and prove that the generated samples are close to this posterior in distribution. However, this does not guarantee that the exact groundtruth sample is recovered. Therefore, motivated by prior works [42, 7], we analyze posterior sampling in a two-step diffusion model and answer a fundamental question: *Can a pre-trained latent diffusion model provably solve inverse problems in a perfectly recoverable setting?*

## 3.2 Posterior Sampling using Latent Diffusion Model

In this section, we analyze two approximations: GML-DPS based on (6), and PSLD based on (7), displayed in **Algorithm** 2. We consider the case where the latent distribution of the VAE is in the same space as the latent distribution of the data generating model, i.e., $k = l$, and normalize $\gamma_i = 1$ (as this is immaterial in the linear setting). In **Proposition 3.3**, we provide analytical solutions for the encoder and the decoder of the VAE.

**Proposition 3.3** (Variational Autoencoder). *Suppose **Assumption 3.1** holds. For an encoder $\mathcal{E} : \mathbb{R}^d \to \mathbb{R}^k$ and a decoder $\mathcal{D} : \mathbb{R}^k \to \mathbb{R}^d$, denote by $\mathcal{L}(\phi, \omega)$ the training objective of VAE:*

$$\arg\min_{\phi, \omega} \mathcal{L}(\phi, \omega) := \mathbb{E}_{\overrightarrow{\boldsymbol{x}_0} \sim p}\left[\left\|\mathcal{D}(\mathcal{E}(\overrightarrow{\boldsymbol{x}_0}; \phi); \omega) - \overrightarrow{\boldsymbol{x}_0}\right\|_2^2\right] + \lambda KL\left(\mathcal{E}\sharp p, \mathcal{N}(\boldsymbol{0}, \boldsymbol{I}_k)\right),$$

*then the combination of $\mathcal{E}(\overrightarrow{\boldsymbol{x}_0}; \phi) = \mathcal{S}^T \overrightarrow{\boldsymbol{x}_0}$ and $\mathcal{D}(\overleftarrow{\boldsymbol{z}_0}; \omega) = \mathcal{S}\overleftarrow{\boldsymbol{z}_0}$ is a minimizer of $\mathcal{L}(\phi, \omega)$.*

Using the encoder $\mathcal{E}(\overrightarrow{\boldsymbol{x}_0}; \phi) = \mathcal{S}^T \overrightarrow{\boldsymbol{x}_0}$, we can use the analytical solution $\boldsymbol{\theta}^*$ of the LDM obtained in **Theorem A.1**. To verify that $\boldsymbol{\theta}^*$ recovers the true subspace $p(\overrightarrow{\boldsymbol{x}_0})$, we compose the decoder $\mathcal{D}(\overleftarrow{\boldsymbol{z}_0}; \omega) = \mathcal{S}\overleftarrow{\boldsymbol{z}_0}$ with the generator of the LDM, i.e., $\overleftarrow{\boldsymbol{x}_0} = \mathcal{D}\left(\boldsymbol{\theta}^* \overleftarrow{\boldsymbol{z}_1}\right) = \mathcal{D}\left(\boldsymbol{I}_k \overleftarrow{\boldsymbol{z}_1}\right) = \mathcal{S}\overleftarrow{\boldsymbol{z}_1}$. Since $\overleftarrow{\boldsymbol{z}_1} \sim \mathcal{N}(\boldsymbol{0}, \boldsymbol{I}_k)$ and $\mathcal{S}$ is the data generating model, this shows that $\overleftarrow{\boldsymbol{x}_0}$ is a sample from $p(\overrightarrow{\boldsymbol{x}_0})$. Thus we have the following.

**Theorem 3.4** (Generative Modeling using Diffusion in Latent Space). *Suppose **Assumption 3.1** holds. Let the optimal solution of the latent diffusion model be*

$$\boldsymbol{\theta}^* = \arg\min_{\boldsymbol{\theta}} \mathbb{E}_{\overrightarrow{\boldsymbol{z}_0}, \overrightarrow{\boldsymbol{\epsilon}}}\left[\left\|\tilde{\mu}_1\left(\overrightarrow{\boldsymbol{z}_1}(\overrightarrow{\boldsymbol{z}_0}, \overrightarrow{\boldsymbol{\epsilon}}), \overrightarrow{\boldsymbol{z}_0}\right) - \mu_\theta\left(\overrightarrow{\boldsymbol{z}_1}\left(\overrightarrow{\boldsymbol{z}_0}, \overrightarrow{\boldsymbol{\epsilon}}\right)\right)\right\|^2\right].$$

*For a fixed variance $\beta > 0$, if $\mu_{\boldsymbol{\theta}}\left(\overrightarrow{\boldsymbol{z}_1}\left(\overrightarrow{\boldsymbol{z}_0}, \overrightarrow{\boldsymbol{\epsilon}}\right)\right) := \boldsymbol{\theta}\overrightarrow{\boldsymbol{z}_1}\left(\overrightarrow{\boldsymbol{z}_0}, \overrightarrow{\boldsymbol{\epsilon}}\right)$, then the closed-form solution is $\boldsymbol{\theta}^* = \sqrt{1 - \beta}\boldsymbol{I}_k$, which after normalization by $\frac{1}{\sqrt{1-\beta}}$ and composition with the decoder $\mathcal{D}\left(\overleftarrow{\boldsymbol{z}_0}; \omega\right) = \mathcal{S}\overleftarrow{\boldsymbol{z}_0}$ recovers the true subspace of $p\left(\overrightarrow{\boldsymbol{x}_0}\right)$.*

With this optimal $\boldsymbol{\theta}^*$, we can now prove exact sample recovery using GML-DPS (6).

**Theorem 3.5** (Posterior Sampling using Goodness Modified Latent DPS). *Let **Assumptions 3.1** and **3.2** hold. Let $\sigma_j, \forall j = 1, \ldots, r$, denote the singular values of $(\mathcal{A}\mathcal{S})^T(\mathcal{A}\mathcal{S})$, and let*

$$\boldsymbol{\theta}^* = \arg\min_{\boldsymbol{\theta}} \mathbb{E}_{\overrightarrow{\boldsymbol{z}_0}, \overrightarrow{\boldsymbol{\epsilon}}}\left[\left\|\tilde{\mu}_1\left(\overrightarrow{\boldsymbol{z}_1}(\overrightarrow{\boldsymbol{z}_0}, \overrightarrow{\boldsymbol{\epsilon}}), \overrightarrow{\boldsymbol{z}_0}\right) - \mu_\theta\left(\overrightarrow{\boldsymbol{z}_1}\left(\overrightarrow{\boldsymbol{z}_0}, \overrightarrow{\boldsymbol{\epsilon}}\right)\right)\right\|^2\right].$$

*Given a partially known image $\overrightarrow{\boldsymbol{x}_0} \sim p(\overrightarrow{\boldsymbol{x}_0})$, any fixed variance $\beta \in (0, 1)$, then with the (unique) step size $\eta_i^j = 1/2\sigma_j, j = 1, 2, \ldots, r$, the GML-DPS Algorithm (6) samples from the true posterior $p(\overrightarrow{\boldsymbol{x}_0}|y)$ and exactly recovers the groundtruth sample, i.e., $\overleftarrow{\boldsymbol{x}_0} = \overrightarrow{\boldsymbol{x}_0}$.*

**Theorem 3.5** shows that GML-DPS (6) recovers the true sample using an LDM. This approach, however, requires the step size $\eta$ to be chosen *coordinate-wise* in a specific manner. Also, multiple natural images could have the same measurements in the pixel space. This is a reasonable concern for

LDMs due to one-to-many mappings of the decoder. Note that the *goodness objective* (Section 2.1) cannot help in this scenario because it assigns uniform probability to many of these latents $\overleftarrow{z_1}$ for which $\nabla_{\overleftarrow{z_1}} \left| \left| \overleftarrow{z_0}(\overleftarrow{z_1})] - \mathcal{E}(\mathcal{D}(\overleftarrow{z_0}(\overleftarrow{z_1}))) \right| \right|^2 = 0$. These challenges motivate the *gluing objective* in **Theorem 3.6**. This is crucial for two reasons. First, we show that it helps recover the true sample even when the step size $\eta$ is chosen arbitrarily. Second, it assigns all the probability mass to the desired (unique) solution in the pixel space.

**Theorem 3.6** (Posterior Sampling using Diffusion in Latent Space). *Let **Assumptions 3.1** and **3.2** hold. Let* $\sigma_j, \forall j = 1, \ldots, r$ *denote the singular values of* $(\mathcal{A}\mathcal{S})^T(\mathcal{A}\mathcal{S})$ *and let*

$$\boldsymbol{\theta}^* = \arg\min_{\boldsymbol{\theta}} \mathbb{E}_{\overrightarrow{z_0}, \overrightarrow{\boldsymbol{\epsilon}}} \left[ \left\| \tilde{\mu}_1 \left( \overrightarrow{z_1}(\overrightarrow{z_0}, \overrightarrow{\boldsymbol{\epsilon}}), \overrightarrow{z_0} \right) - \mu_{\boldsymbol{\theta}} \left( \overrightarrow{z_1} \left( \overrightarrow{z_0}, \overrightarrow{\boldsymbol{\epsilon}} \right) \right) \right\|^2 \right].$$

*Given a partially known image* $\overrightarrow{\boldsymbol{x}_0} \sim p(\overrightarrow{\boldsymbol{x}_0})$, *any fixed variance* $\beta \in (0, 1)$, *and any positive step sizes* $\eta_i^j, j = 1, 2, \ldots, r$, *the PSLD Algorithm 2 samples from the true posterior* $p(\overrightarrow{\boldsymbol{x}_0}|y)$ *and exactly recovers the groundtruth sample, i.e.,* $\overleftarrow{\boldsymbol{x}_0} = \overrightarrow{\boldsymbol{x}_0}$.

The important distinction between **Theorem 3.5** and **Theorem 3.6** is that the former requires the *exact* step size while the latter works for any finite step size. Combining denoising, measurement-consistency (with a scalar $\eta$), and gluing updates, we have

$$\overleftarrow{z_0} = \boldsymbol{\theta}^* \overleftarrow{z_1} - \eta \nabla_{\overleftarrow{z_1}} \left\| \mathcal{A}\mathcal{D}(\overleftarrow{z_0}(\overleftarrow{z_1})) - \boldsymbol{y} \right\|_2^2 - \nabla_{\overleftarrow{z_1}} \left\| \overleftarrow{z_0}(\overleftarrow{z_1}) - \mathcal{E}(\mathcal{A}^T \mathcal{A}\overrightarrow{\boldsymbol{x}_0} + (\boldsymbol{I}_d - \mathcal{A}^T \mathcal{A})\mathcal{D}(\overleftarrow{z_0}(\overleftarrow{z_1}))) \right\|_2^2.$$

When $\eta$ is chosen arbitrarily, then the third term guides the reverse SDE towards the optimal solution $\overrightarrow{z_0}$. When the reverse SDE generates the exact same groundtruth sample, i.e., $\mathcal{D}(\overleftarrow{z_1}(\overleftarrow{z_0})) = \overrightarrow{\boldsymbol{x}_0}$, then the third term becomes zero. For all other samples, it penalizes the reverse SDE. Thus, it forces the reverse SDE to recover the true underlying sample irrespective of the value of $\eta$.

We draw the following key insights from our **Theorem 3.6**: **Curse of ambient dimension:** In order to run posterior sampling using diffusion in the pixel space, the gradient of the measurement error needs to be computed in the $d$-dimensional ambient space. Therefore, DPS algorithm suffers from the curse of ambient dimension. On the other hand, our algorithm uses diffusion in the latent space, and therefore avoids the curse of ambient dimension. **Large-scale foundation model:** We propose a posterior sampling algorithm which offers the provision to use large-scale foundation models, and it provably solves general linear inverse problems. **Robustness to measurement step:** The gluing objective makes our algorithm robust to the choice of step size $\eta$. Furthermore, it allows the same (scalar) step size across all the coordinates of $\overrightarrow{\boldsymbol{x}_0}$.

## 4 Experimental Evaluation

We experiment with in-distribution and out-of-distribution datasets. For in-distribution, we conduct our experiments on a subset of the FFHQ dataset [25] (downscaled to $256 \times 256$[3], denoted by FFHQ 256). For out-of-distribution, we use images from the web and ImageNet dataset [17] (resized to $256 \times 256$, denoted by ImageNet 256). To make a fair comparison, we use the same validation subset and follow the same masking strategy as the baseline DPS [11]. It is important to note that our main contribution is an algorithm that can leverage any latent diffusion model. We

Table 1: Quantitative super-resolution (using measurement operator from [32]) results on FFHQ 256 validation samples [25, 11]. We use PSLD with Stable Diffusion. Table shows LPIPS ($\downarrow$).

| Method | PSLD (Ours) | DPS [11] |
|--------|-------------|----------|
| $2\times$ | **0.185** | 0.220 |
| $3\times$ | **0.220** | 0.247 |
| $4\times$ | **0.233** | 0.291 |

test our algorithm with two pre-trained latent diffusion models: (i) the Stable Diffusion model that is trained on multiple subsets of the LAION dataset [44, 45]; and (ii) the Latent Diffusion model (LDM-VQ-4) trained on the FFHQ 256 dataset [41]. The DPS model is similarly trained from scratch for 1M steps using 49k FFHQ 256 images, which excludes the first 1K images used as validation set.

**Inverse Problems.** We experiment with the following task-specific measurement operators from the baseline DPS [11]: (i) Box inpainting uses a mask of size 128×128 at the center. (ii) Random inpainting chooses a drop probability uniformly at random between $(0.2, 0.8)$ and applies this drop

---

[3]https://www.kaggle.com/datasets/denislukovnikov/ffhq256-images-only

Table 2: Quantitative inpainting results on FFHQ 256 validation set [25, 11]. We use Stable Diffusion v-1.5 and the measurement operators as in DPS [11]. As shown, our PSLD model outperforms DPS since it is able to leverage the power of the Stable Diffusion foundation model.

| | Inpaint (random) | | Inpaint (box) | | SR ($4\times$) | | Gaussian Deblur | |
| --- | --- | --- | --- | --- | --- | --- | --- | --- |
| Method | FID ($\downarrow$) | LPIPS ($\downarrow$) | FID ($\downarrow$) | LPIPS ($\downarrow$) | FID ($\downarrow$) | LPIPS ($\downarrow$) | FID ($\downarrow$) | LPIPS ($\downarrow$) |
| PSLD (Ours) | **21.34** | **0.096** | 43.11 | **0.167** | **34.28** | **0.201** | **41.53** | **0.221** |
| DPS [11] | 33.48 | 0.212 | **35.14** | 0.216 | 39.35 | 0.214 | 44.05 | 0.257 |
| DDRM [26] | 69.71 | 0.587 | 42.93 | 0.204 | 62.15 | 0.294 | 74.92 | 0.332 |
| MCG [13] | 29.26 | 0.286 | 40.11 | 0.309 | 87.64 | 0.520 | 101.2 | 0.340 |
| PnP-ADMM [6] | 123.6 | 0.692 | 151.9 | 0.406 | 66.52 | 0.353 | 90.42 | 0.441 |
| Score-SDE [50] | 76.54 | 0.612 | 60.06 | 0.331 | 96.72 | 0.563 | 109.0 | 0.403 |
| ADMM-TV | 181.5 | 0.463 | 68.94 | 0.322 | 110.6 | 0.428 | 186.7 | 0.507 |

probability to all the pixels. (iii) Super-resolution downsamples images at $4\times$ scale. (iv) Gaussian blur convolves images with a Gaussian blur kernel. (v) Motion blur convolves images with a motion blur kernel. We also experiment with these additional operators from RePaint [32]: (vi) Super-resolution downsamples images at $2\times$, $3\times$, and $4\times$ scale. (vii) Denoising has Gaussian noise with $\sigma = 0.05$. (viii) Destriping has vertical and horizontal stripes in the input images.

**Evaluation.** We compare the performance of our PSLD algorithm with the state-of-the-art DPS algorithm [11] on random inpainting, box inpainting, denoising, Gaussian deblur, motion deblur, arbitrary masking, and super-resolution tasks. We show that PSLD outperforms DPS, both in-distribution and out-of-distribution datasets, using the Stable Diffusion v-1.5 model pre-trained on the LAION dataset. We also test PSLD with LDM-VQ-4 trained on FFHQ 256, to compare with DPS trained on the same data distribution. Note that the LDM-v4 is a latent-based model released prior to Stable Diffusion. Therefore, it does not match the performance of Stable Diffusion in solving inverse problems. However, it shows the general applicability of our framework to leverage an LDM in posterior sampling. Since Stable Diffusion v-1.5 is trained with an image resolution of $512 \times 512$, we apply the forward operator after upsampling inputs to $512 \times 512$, run posterior sampling at $512 \times 512$, and then downsample images to the original $256 \times 256$ resolution for a fair comparison with DPS. We observed a similar performance while applying the masking operator at $256 \times 256$ and upscaling to $512 \times 512$ before running PSLD. More implementation details are provided in Appendix C.1.

**Metrics.** We use the commonly used Learned Perceptual Image Patch Similarity (LPIPS), Peak Signal-to-Noise Ratio (PSNR), Structural Similarity Index Metric (SSIM), and Fréchet Inception Distance[4] (FID) metrics for quantitative evaluation.

**Results.** Figure 2 shows the inpainting results on out-of-distribution samples. This experiment was performed on commercial platforms that use (to the best of our knowledge) Stable diffusion and additional proprietary models. This evaluation was performed on models deployed in May 2023 and may change as commercial providers improve their platforms.

The qualitative advantage of PSLD is clearly demonstrated in Figures 2, 3, 4, 15 and 16. In Figure 5, we compare PSLD and DPS in random inpainting task for varying percentage of dropped pixels. Quantitatively, PSLD outperforms DPS in commonly used metrics: LPIPS, PSNR, and SSIM.

In our PSLD algorithm, we use Stable Diffusion v1.5 model and (zero-shot) test it on inverse problems. Table 6 compares the quantitative results of PSLD with related works on random inpainting, box inpainting, super-resolution, and Gaussian deblur tasks. PSLD significantly outperforms previous approaches on the relatively easier random inpainting task, and it is better or comparable on harder tasks. Table 4 draws a comparison between PSLD and the strongest baseline (among the compared methods) on out-of-distribution images. Table 1 shows the super-resolution results using nearest-neighbor kernels from [32] on FFHQ 256 validation dataset. Observe that PSLD outperforms state-of-the-art methods across diverse tasks and standard evaluation metrics.

In Table 3, we compare PSLD (using LDM-VQ-4) and DPS on random and box inpainting tasks with the same operating resolution ($256 \times 256$) and training distributions (FFHQ 256). Although the LDM model exceeds DPS performance in box inpainting, it is comparable in random inpainting. As expected, using a more powerful pre-trained model such as Stable Diffusion is beneficial in

---

[4]https://github.com/mseitzer/pytorch-fid

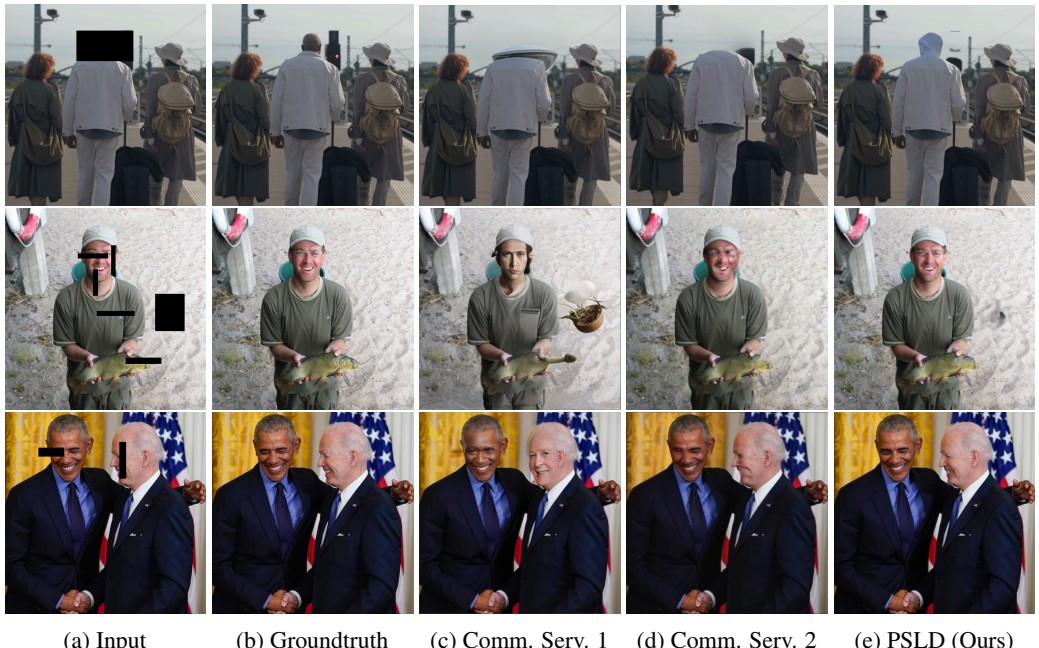

| (a) Input | (b) Groundtruth | (c) Comm. Serv. 1 | (d) Comm. Serv. 2 | (e) PSLD (Ours) |

Figure 2: Inpainting results in general domain images from the web (see Appendix C for image sources). Our model compared to state-of-art commercial inpainting services that leverage the same foundation model (Stable Diffusion v-1.5).

Table 3: Quantitative inpainting results on FFHQ 256 validation set [25, 11]. We use the *latent diffusion* (LDM-VQ-4) trained on FFHQ 256. Note that in this experiment PSLD and DPS use diffusion models trained on the same dataset. As shown, PSLD with LDM-VQ-4 as diffusion model outperforms DPS in box inpainting and has comparable performance in random inpainting.

| | Inpaint (random) | | | Inpaint (box) | | |
|---|---|---|---|---|---|---|
| Method | PSNR ($\uparrow$) | SSIM ($\uparrow$) | LPIPS ($\downarrow$) | PSNR ($\uparrow$) | SSIM ($\uparrow$) | LPIPS ($\downarrow$) |
| PSLD (Ours) | **30.31** | **0.851** | 0.221 | **24.22** | **0.819** | **0.158** |
| DPS [11] | 29.49 | 0.844 | **0.212** | 23.39 | 0.798 | 0.214 |

Table 4: Quantitative results of random inpainting and denoising on FFHQ 256 [25, 11] using Stable Diffusion v-1.5. Note that DPS is trained on FFHQ 256. The results show that our method PSLD generalizes well to out-of-distribution samples even without finetuning.

| | Random inpaint + denoise $\sigma = 0.00$ | | | Random inpaint + denoise $\sigma = 0.05$ | | |
|---|---|---|---|---|---|---|
| Method | PSNR ($\uparrow$) | SSIM ($\uparrow$) | LPIPS ($\downarrow$) | PSNR ($\uparrow$) | SSIM ($\uparrow$) | LPIPS ($\downarrow$) |
| PSLD (Ours) | **34.02** | **0.951** | **0.083** | **33.71** | **0.943** | **0.096** |
| DPS [11] | 31.41 | 0.884 | 0.171 | 29.49 | 0.844 | 0.212 |

reconstruction–see Table 6. This highlights the significance of our PSLD algorithm that has the provision to incorporate a powerful foundation model with no extra training costs for solving inverse problems. Importantly, PSLD uses latent-based diffusion, and thus it avoids the curse of ambient dimension (**Theorem 3.6**), while still achieving comparable results to the state-of-the-art method DPS [11] that has been trained on the same dataset. Additional experimental evaluation is provided in Appendix C.

## 5 Conclusion

In this paper, we leverage latent diffusion models to solve general linear inverse problems. While previously proposed approaches only apply to pixel-space diffusion models, our algorithm allows us to use the image prior learned by latent-based foundation generative models. We provide a principled analysis of our algorithm in a linear two-step diffusion setting, and use insights from this analysis to design a modified objective (goodness and gluing). This leads to our algorithm – Posterior

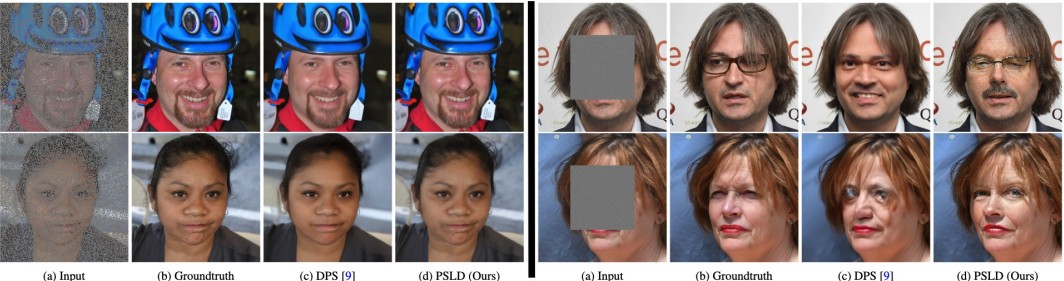

Figure 3: **Left panel:** Random Inpainting on images from FFHQ 256 [25] using PSLD with Stable Diffusion v-1.5. Notice the text in the top row and the facial expression in the bottom row. **Right panel:** Block $(128 \times 128)$ inpainting, using the LDM-VQ-4 model trained on FFHQ 256 [25]. Notice the glasses in the top row and eyes in the bottom row.

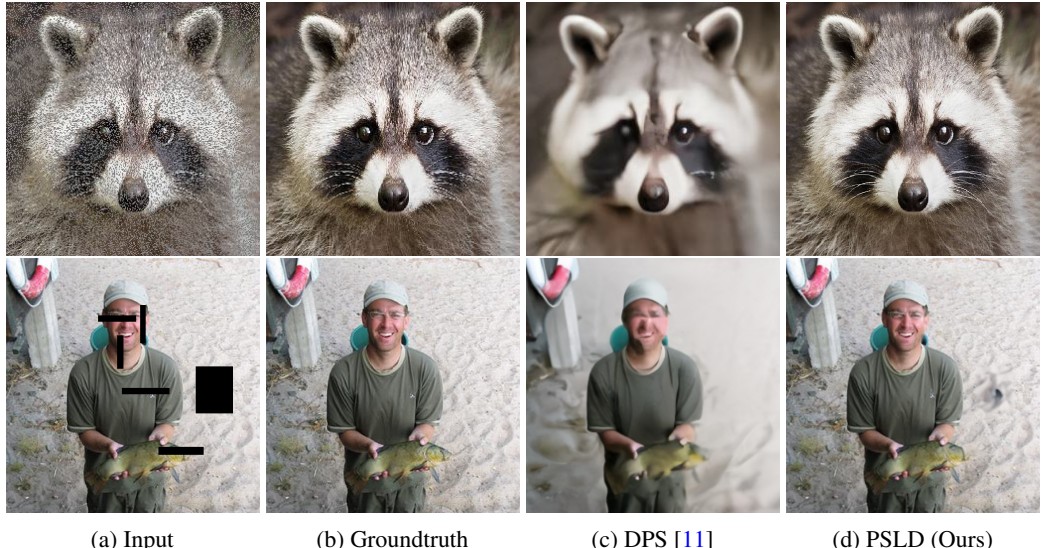

    (a) Input         (b) Groundtruth         (c) DPS [11]         (d) PSLD (Ours)

Figure 4: Inpainting (random and box) results on out-of-distribution samples, $256 \times 256$ (see Appendix C for image sources). We use PSLD with Stable Diffusion v-1.5 as generative foundation model.

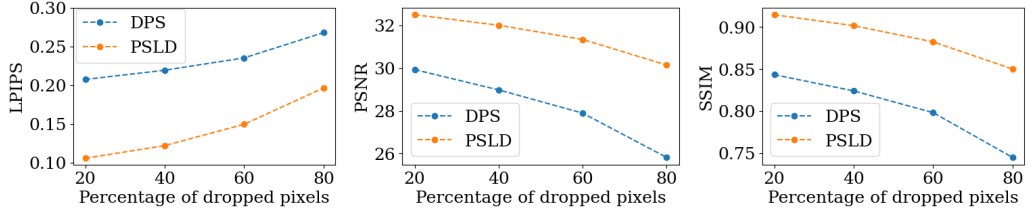

Figure 5: Comparing DPS and PSLD performance in random inpainting on FFHQ 256 [25, 11], as the percentage of masked pixels increases. PSLD with Stable Diffusion outperforms DPS.

Sampling with Latent Diffusion (PSLD) – that experimentally outperforms state-of-art baselines on a wide variety of tasks including random inpainting, block inpainting, denoising, destriping, and super-resolution.

**Limitations.** Our evaluation is based on Stable Diffusion which was trained on the LAION dataset. Biases in this dataset and foundation model will be implicitly affecting our algorithm. Our method can work with any LDM and we expect new foundation models trained on better datasets like [19] to mitigate these issues. Second, we have not explored how to use latent-based foundation models to solve non-linear inverse problems. Our method builds on the DPS approximation (which performs well on non-linear inverse problems), and hence we believe our method can also be similarly extended.

## Acknowledgements

This research has been supported by NSF Grants 2019844, 2112471, AF 1901292, CNS 2148141, Tripods CCF 1934932, the Texas Advanced Computing Center (TACC) and research gifts by Western Digital, Wireless Networking and Communications Group (WNCG) Industrial Affiliates Program, UT Austin Machine Learning Lab (MLL), Cisco and the Stanly P. Finch Centennial Professorship in Engineering. Litu Rout has been supported by the Ju-Nam and Pearl Chew Endowed Presidential Fellowship in Engineering. Giannis Daras has been supported by the Onassis Fellowship (Scholarship ID: F ZS 012-1/2022-2023), the Bodossaki Fellowship and the Leventis Fellowship. We thank the HuggingFace team for providing us GPU support for the demo of our work.

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

# A  Additional Theoretical Results

**Notation and Measurement Matrix.** We elaborate on the structure of the measurement matrix $\mathcal{A} \in \mathbb{R}^{l \times d}$. In our setting, we are considering linear inverse problems. Thus, this matrix is a pixel selector and consists of a subset of the rows from the $d \times d$ identity matrix (the rows that are present correspond to the indices of the selected pixels from the image $\overrightarrow{\boldsymbol{x}_0} \in \mathbb{R}^d$). Given this structure, it immediately follows that $\mathcal{A}^T \mathcal{A}$ is a $d \times d$ matrix that has the interpretation of a pixel selection *mask*. Specifically, $\mathcal{A}^T \mathcal{A}$ is a $d \times d$ diagonal matrix $\boldsymbol{D}(\boldsymbol{m})$, where the elements of $\boldsymbol{m}$ are set to 1 where data (pixel) is observed and 0 where data (pixel) is masked. Without the loss of generality, we suppose that the first $k$ coordinates are known.

## A.1  Posterior Sampling using Pixel-space Diffusion Model

We first consider the reverse process, starting with $\overleftarrow{\boldsymbol{x}_1} \sim \mathcal{N}\left(\boldsymbol{0}, \boldsymbol{I}_d\right)$, and borrow a result from [42] to show that the sample $\overleftarrow{\boldsymbol{x}_0}$ generated by the reverse process is a valid image from $p(\overrightarrow{\boldsymbol{x}_0})$.

**Theorem A.1** (Generative Modeling using Diffusion in Pixel Space, [42]). *Suppose **Assumption 3.1** holds. Let*

$$\boldsymbol{\theta}^* = \arg\min_{\boldsymbol{\theta}} \mathbb{E}_{\overrightarrow{\boldsymbol{x}_0}, \overrightarrow{\boldsymbol{\epsilon}}} \left[ \left\| \tilde{\mu}_1\left(\overrightarrow{\boldsymbol{x}_1}(\overrightarrow{\boldsymbol{x}_0}, \overrightarrow{\boldsymbol{\epsilon}}), \overrightarrow{\boldsymbol{x}_0}\right) - \mu_{\boldsymbol{\theta}}\left(\overrightarrow{\boldsymbol{x}_1}\left(\overrightarrow{\boldsymbol{x}_0}, \overrightarrow{\boldsymbol{\epsilon}}\right)\right) \right\|^2 \right].$$

*For a fixed variance $\beta > 0$, if $\mu_{\boldsymbol{\theta}}\left(\overrightarrow{\boldsymbol{x}_1}\left(\overrightarrow{\boldsymbol{x}_0}, \overrightarrow{\boldsymbol{\epsilon}}\right)\right) := \boldsymbol{\theta} \overrightarrow{\boldsymbol{x}_1}\left(\overrightarrow{\boldsymbol{x}_0}, \overrightarrow{\boldsymbol{\epsilon}}\right)$, then the closed-form solution $\boldsymbol{\theta}^*$ is $\sqrt{1 - \beta} \boldsymbol{S} \boldsymbol{S}^T$, which after normalization by $1/\sqrt{1 - \beta}$ recovers the true subspace of $p\left(\overrightarrow{\boldsymbol{x}_0}\right)$.*

Though this establishes that $\overleftarrow{\boldsymbol{x}_0}$ generated by the reverse process is a valid image from $p(\overrightarrow{\boldsymbol{x}_0})$, it is not necessarily a sample from the posterior $p(\overrightarrow{\boldsymbol{x}_0}|\boldsymbol{y})$ that satisfies the measurements. To accomplish this we perform one additional step of gradient descent for every step of the reverse process. This gives us **Algorithm 1**, the DPS algorithm. The next theorem shows that the reverse SDE guided by these measurements (3) recovers the true underlying sample[5].

**Theorem A.2** (Posterior Sampling using Diffusion in Pixel Space). *Suppose **Assumption 3.1** and **Assumption 3.2** hold. Let us denote by $\sigma_j, \forall j = 1, \ldots, r$, the singular values of $(\mathcal{A}\mathcal{S})^T(\mathcal{A}\mathcal{S})$ and*

$$\boldsymbol{\theta}^* = \arg\min_{\boldsymbol{\theta}} \mathbb{E}_{\overrightarrow{\boldsymbol{x}_0}, \overrightarrow{\boldsymbol{\epsilon}}} \left[ \left\| \tilde{\mu}_1\left(\overrightarrow{\boldsymbol{x}_1}(\overrightarrow{\boldsymbol{x}_0}, \overrightarrow{\boldsymbol{\epsilon}}), \overrightarrow{\boldsymbol{x}_0}\right) - \mu_{\boldsymbol{\theta}}\left(\overrightarrow{\boldsymbol{x}_1}\left(\overrightarrow{\boldsymbol{x}_0}, \overrightarrow{\boldsymbol{\epsilon}}\right)\right) \right\|^2 \right].$$

*Given a partially known image $\overrightarrow{\boldsymbol{x}_0} \sim p(\overrightarrow{\boldsymbol{x}_0})$, a fixed variance $\beta > 0$, there exists a step size $\zeta_i^j = 1/2\sigma_j$ for all the coordinates of $\overrightarrow{\boldsymbol{x}_0}$ such that **Algorithm 1** samples from the true posterior $p(\overrightarrow{\boldsymbol{x}_0}|y)$ and exactly recovers the groundtruth sample, i.e., $\overleftarrow{\boldsymbol{x}_0} = \overrightarrow{\boldsymbol{x}_0}$.*

# B  Technical Proofs

This section contains proofs of all the theorems and propositions presented in the main body of the paper. For clarity, we restate the theorems more formally with precise mathematical details.

## B.1  Proof of Theorem A.2

**Theorem B.1** (Posterior Sampling using Diffusion in Pixel Space). *Suppose **Assumption 3.1** and **Assumption 3.2** hold. Let us denote by $\boldsymbol{\sigma} = \{\sigma_j\}_{j=1}^k$ the singular values of $(\mathcal{A}\mathcal{S})^T(\mathcal{A}\mathcal{S})$, i.e. $(\mathcal{A}\mathcal{S})^T(\mathcal{A}\mathcal{S}) = \boldsymbol{U}\boldsymbol{\Sigma}\boldsymbol{V}^T := \boldsymbol{U}\boldsymbol{D}(\boldsymbol{\sigma})\boldsymbol{V}^T, \boldsymbol{U} \in \mathbb{R}^{k \times k}, \boldsymbol{V} \in \mathbb{R}^{k \times k}$ and*

$$\boldsymbol{\theta}^* = \arg\min_{\boldsymbol{\theta}} \mathbb{E}_{\overrightarrow{\boldsymbol{x}_0}, \overrightarrow{\boldsymbol{\epsilon}}} \left[ \left\| \tilde{\mu}_1\left(\overrightarrow{\boldsymbol{x}_1}(\overrightarrow{\boldsymbol{x}_0}, \overrightarrow{\boldsymbol{\epsilon}}), \overrightarrow{\boldsymbol{x}_0}\right) - \mu_{\theta}\left(\overrightarrow{\boldsymbol{x}_1}\left(\overrightarrow{\boldsymbol{x}_0}, \overrightarrow{\boldsymbol{\epsilon}}\right)\right) \right\|^2 \right].$$

---

[5]While the DPS Algorithm [11] uses a scalar step size $\zeta_i$ at each step, this does not suffice for exact recovery. However, by generalizing to allow a different step size per coordinate, we can show sample recovery. Thus, in this section, we denote $\zeta_i^j$ to be the step size at step $i$ and coordinate $j$, $1 \leq j \leq r$. Also note that the step index $i$ is vacuous in this section, as we consider a two-step diffusion process (i.e., $i$ is always '1').

Suppose $\overrightarrow{\boldsymbol{x}_0} \sim p(\overrightarrow{\boldsymbol{x}_0})$. *Given measurements* $y = \mathcal{A}\overrightarrow{\boldsymbol{x}_0}$ *and a fixed variance* $\beta \in (0, 1)$, *there exists a matrix step size*[6] $\boldsymbol{\zeta} = (1/2)(\mathcal{S}\boldsymbol{U})\boldsymbol{D}(\boldsymbol{\zeta}_i)(\mathcal{S}\boldsymbol{U})^T, \boldsymbol{\zeta}_i = \{\zeta_i^j = 1/\sigma_j\}_{j=1}^k$ *for all the coordinates of* $\overrightarrow{\boldsymbol{x}_0}$ *such that **Algorithm 1** samples from the true posterior* $p(\overrightarrow{\boldsymbol{x}_0}|y)$ *and exactly recovers the groundtruth sample, i.e.,* $\overleftarrow{\boldsymbol{x}_0} = \overrightarrow{\boldsymbol{x}_0}$.

*Proof.* Our goal is to show that $\overleftarrow{\boldsymbol{x}_0} = \overrightarrow{\boldsymbol{x}_0}$, where $\overleftarrow{\boldsymbol{x}_0}$ is returned by **Algorithm 1**. Recall that the reverse process starts with $\overleftarrow{\boldsymbol{x}_1} \sim \mathcal{N}(\boldsymbol{0}, \boldsymbol{I}_d)$ and generates the following:

$$
\begin{aligned}
\overleftarrow{\boldsymbol{x}_0} &= \boldsymbol{\theta}^* \overleftarrow{\boldsymbol{x}_1} - \boldsymbol{\zeta} \nabla_{\overleftarrow{\boldsymbol{x}_1}} \left\| \mathcal{A}\overleftarrow{\boldsymbol{x}_0}(\overleftarrow{\boldsymbol{x}_1}) - \boldsymbol{y} \right\|_2^2 \\
&= \boldsymbol{\theta}^* \overleftarrow{\boldsymbol{x}_1} - \boldsymbol{\zeta} \nabla_{\overleftarrow{\boldsymbol{x}_1}} \left\| \mathcal{A}\mathcal{S}\mathcal{S}^T \overleftarrow{\boldsymbol{x}_1} - \boldsymbol{y} \right\|_2^2 \\
&= \mathcal{S}\mathcal{S}^T \overleftarrow{\boldsymbol{x}_1} - 2\boldsymbol{\zeta} \left( \mathcal{A}\mathcal{S}\mathcal{S}^T \right)^T \left( \mathcal{A}\mathcal{S}\mathcal{S}^T \overleftarrow{\boldsymbol{x}_1} - \boldsymbol{y} \right) \\
&= \mathcal{S}\mathcal{S}^T \overleftarrow{\boldsymbol{x}_1} - 2\boldsymbol{\zeta}\mathcal{S}\mathcal{S}^T \mathcal{A}^T \left( \mathcal{A}\mathcal{S}\mathcal{S}^T \overleftarrow{\boldsymbol{x}_1} - \boldsymbol{y} \right) \\
&= \mathcal{S}\mathcal{S}^T \overleftarrow{\boldsymbol{x}_1} - 2\boldsymbol{\zeta}\mathcal{S}\mathcal{S}^T \mathcal{A}^T \mathcal{A}\mathcal{S}\mathcal{S}^T \overleftarrow{\boldsymbol{x}_1} + 2\boldsymbol{\zeta}\mathcal{S}\mathcal{S}^T \mathcal{A}^T \boldsymbol{y} \\
&= \mathcal{S}\mathcal{S}^T \overleftarrow{\boldsymbol{x}_1} - 2\boldsymbol{\zeta}\mathcal{S}\mathcal{S}^T \mathcal{A}^T \mathcal{A}\mathcal{S}\mathcal{S}^T \overleftarrow{\boldsymbol{x}_1} + 2\boldsymbol{\zeta}\mathcal{S}\mathcal{S}^T \mathcal{A}^T \mathcal{A}\overrightarrow{\boldsymbol{x}_0} \\
&= \mathcal{S}\mathcal{S}^T \overleftarrow{\boldsymbol{x}_1} - 2\boldsymbol{\zeta}\mathcal{S}\mathcal{S}^T \mathcal{A}^T \mathcal{A}\mathcal{S}\mathcal{S}^T \overleftarrow{\boldsymbol{x}_1} + 2\boldsymbol{\zeta}\mathcal{S}\mathcal{S}^T \mathcal{A}^T \mathcal{A}\mathcal{S}\overrightarrow{\boldsymbol{z}_0}.
\end{aligned}
$$

Now, we use the singular value decomposition of $(\mathcal{A}\mathcal{S})^T(\mathcal{A}\mathcal{S})$ with left singular vectors in $\boldsymbol{U} \in \mathbb{R}^{k \times k}$, right singular vectors in $\boldsymbol{V} \in \mathbb{R}^{k \times k}$, and singular values $\boldsymbol{\sigma} = [\sigma_1, \ldots, \sigma_k]$ in $\Sigma = \boldsymbol{D}(\boldsymbol{\sigma})$. Thus, the above expression becomes

$$
\begin{aligned}
\overleftarrow{\boldsymbol{x}_0} &= \mathcal{S}\mathcal{S}^T \overleftarrow{\boldsymbol{x}_1} - 2\boldsymbol{\zeta}\mathcal{S}\boldsymbol{U}\Sigma\boldsymbol{V}^T \boldsymbol{S}^T \overleftarrow{\boldsymbol{x}_1} + 2\boldsymbol{\zeta}\mathcal{S}\boldsymbol{U}\Sigma\boldsymbol{V}^T \overrightarrow{\boldsymbol{z}_0} \\
&= \mathcal{S}\mathcal{S}^T \overleftarrow{\boldsymbol{x}_1} - 2\boldsymbol{\zeta}\mathcal{S}\boldsymbol{U}\Sigma\boldsymbol{V}^T \boldsymbol{S}^T \overleftarrow{\boldsymbol{x}_1} + 2\boldsymbol{\zeta}\mathcal{S}\boldsymbol{U}\Sigma\boldsymbol{V}^T \overrightarrow{\boldsymbol{z}_0} \\
&= \mathcal{S}\mathcal{S}^T \overleftarrow{\boldsymbol{x}_1} - 2(\mathcal{S}\boldsymbol{U})\boldsymbol{D}(\boldsymbol{\zeta}_i)(\mathcal{S}\boldsymbol{U})^T \mathcal{S}\boldsymbol{U}\Sigma\boldsymbol{V}^T \boldsymbol{S}^T \overleftarrow{\boldsymbol{x}_1} + 2(\mathcal{S}\boldsymbol{U})\boldsymbol{D}(\boldsymbol{\zeta}_i)(\mathcal{S}\boldsymbol{U})^T \mathcal{S}\boldsymbol{U}\Sigma\boldsymbol{V}^T \overrightarrow{\boldsymbol{z}_0} \\
&\overset{(i)}{=} \mathcal{S}\mathcal{S}^T \overleftarrow{\boldsymbol{x}_1} - 2(\mathcal{S}\boldsymbol{U})\boldsymbol{D}(\boldsymbol{\zeta}_i)\boldsymbol{U}^T \boldsymbol{S}^T \mathcal{S}\boldsymbol{U}\Sigma\boldsymbol{V}^T \boldsymbol{S}^T \overleftarrow{\boldsymbol{x}_1} + 2(\mathcal{S}\boldsymbol{U})\boldsymbol{D}(\boldsymbol{\zeta}_i)\boldsymbol{U}^T \boldsymbol{S}^T \mathcal{S}\boldsymbol{U}\Sigma\boldsymbol{V}^T \overrightarrow{\boldsymbol{z}_0} \\
&\overset{(ii)}{=} \mathcal{S}\mathcal{S}^T \overleftarrow{\boldsymbol{x}_1} - 2(\mathcal{S}\boldsymbol{U})\boldsymbol{D}(\boldsymbol{\zeta}_i)\boldsymbol{U}^T \boldsymbol{U}\Sigma\boldsymbol{U}^T \boldsymbol{S}^T \overleftarrow{\boldsymbol{x}_1} + 2(\mathcal{S}\boldsymbol{U})\boldsymbol{D}(\boldsymbol{\zeta}_i)\boldsymbol{U}^T \boldsymbol{U}\Sigma\boldsymbol{U}^T \overrightarrow{\boldsymbol{z}_0} \\
&= \mathcal{S}\mathcal{S}^T \overleftarrow{\boldsymbol{x}_1} - 2(\mathcal{S}\boldsymbol{U})\boldsymbol{D}(\boldsymbol{\zeta}_i)\Sigma\boldsymbol{U}^T \boldsymbol{S}^T \overleftarrow{\boldsymbol{x}_1} + 2(\mathcal{S}\boldsymbol{U})\boldsymbol{D}(\boldsymbol{\zeta}_i)\Sigma\boldsymbol{U}^T \overrightarrow{\boldsymbol{z}_0} \\
&= \mathcal{S}\mathcal{S}^T \overleftarrow{\boldsymbol{x}_1} - 2\mathcal{S}\boldsymbol{U}\boldsymbol{D}(\boldsymbol{\zeta}_i)\boldsymbol{D}(\boldsymbol{\sigma})\boldsymbol{U}^T \boldsymbol{S}^T \overleftarrow{\boldsymbol{x}_1} + 2\mathcal{S}\boldsymbol{U}\boldsymbol{D}(\boldsymbol{\zeta}_i)\boldsymbol{D}(\boldsymbol{\sigma})\boldsymbol{U}^T \overrightarrow{\boldsymbol{z}_0} \\
&= \mathcal{S}\mathcal{S}^T \overleftarrow{\boldsymbol{x}_1} - 2\mathcal{S}\boldsymbol{U}\boldsymbol{D}(\boldsymbol{\zeta}_i \odot \boldsymbol{\sigma})\boldsymbol{U}^T \boldsymbol{S}^T \overleftarrow{\boldsymbol{x}_1} + 2\mathcal{S}\boldsymbol{U}\boldsymbol{D}(\boldsymbol{\zeta}_i \odot \boldsymbol{\sigma})\boldsymbol{U}^T \overrightarrow{\boldsymbol{z}_0},
\end{aligned}
$$

where (i) is due to **Assumption 3.1** and (ii) uses **Assumption 3.2**. By choosing $\zeta_i^j$ as half the inverse of the non-zero singular values of $(\mathcal{A}\mathcal{S})^T(\mathcal{A}\mathcal{S})$, i.e., $\zeta_i^j = 1/2\sigma_i \ \forall i = 1, \ldots, k$, we obtain

$$
\begin{aligned}
\overleftarrow{\boldsymbol{x}_0} &= \mathcal{S}\mathcal{S}^T \overleftarrow{\boldsymbol{x}_1} - \mathcal{S}\boldsymbol{U}\boldsymbol{U}^T \boldsymbol{S}^T \overleftarrow{\boldsymbol{x}_1} + \mathcal{S}\boldsymbol{U}\boldsymbol{U}^T \overrightarrow{\boldsymbol{z}_0} \\
&= \mathcal{S}\mathcal{S}^T \overleftarrow{\boldsymbol{x}_1} - \mathcal{S}\mathcal{S}^T \overleftarrow{\boldsymbol{x}_1} + \mathcal{S}\overrightarrow{\boldsymbol{z}_0} = \overrightarrow{\boldsymbol{x}_0},
\end{aligned}
$$

which completes the statement of the theorem. $\qquad\square$

## B.2 Proof of Proposition 3.3

**Proposition B.2** (Variational Autoencoder). *Suppose **Assumption 3.1** holds. For an encoder* $\mathcal{E} : \mathbb{R}^d \to \mathbb{R}^k$ *and a decoder* $\mathcal{D} : \mathbb{R}^k \to \mathbb{R}^d$, *denote by* $\mathcal{L}(\phi, \omega)$ *the training objective of VAE:*

$$
\arg\min_{\phi, \omega} \mathcal{L}(\phi, \omega) := \mathbb{E}_{\overrightarrow{\boldsymbol{x}_0} \sim p} \left[ \left\| \mathcal{D}(\mathcal{E}(\overrightarrow{\boldsymbol{x}_0}; \phi); \omega) - \overrightarrow{\boldsymbol{x}_0} \right\|_2^2 \right] + \lambda KL\left( \mathcal{E}\sharp p, \mathcal{N}(\boldsymbol{0}, \boldsymbol{I}_k) \right),
$$

*then the combination of* $\mathcal{E}(\overrightarrow{\boldsymbol{x}_0}; \phi) = \mathcal{S}^T \overrightarrow{\boldsymbol{x}_0}$ *and* $\mathcal{D}(\overleftarrow{\boldsymbol{z}_0}; \omega) = \mathcal{S}\overleftarrow{\boldsymbol{z}_0}$ *is a minimizer of* $\mathcal{L}(\phi, \omega)$.

---

[6]We use the term 'step size' in a more general way than is normally used. In this case, the step size is a 'pre-conditioning' positive definite matrix, whose eigenvalue magnitudes correspond to the scalar step sizes per coordinate along an appropriately rotated basis. This general form is needed and with carefully selected (unique) eigenvalues; otherwise the DPS algorithm fails to converge to the groundtruth sample. We will later see that for our PSLD Algorithm in Theorem 3.6, we can revert to the commonly used notion of step size (a single scalar), as any finite step size (including a single scalar common across all coordinates) suffices for proving recovery.

*Proof.* To show that the encoder $\mathcal{E}(\overrightarrow{\boldsymbol{x}_0}; \phi) = \mathcal{S}^T \overrightarrow{\boldsymbol{x}_0}$ and the decoder $\mathcal{D}(\overleftarrow{\boldsymbol{z}_0}; \omega) = \mathcal{S}\overleftarrow{\boldsymbol{z}_0}$ minimize the VAE training objective $\mathcal{L}(\phi, \omega)$, we begin with the first part of the loss, which is also called *reconstruction error* $\mathcal{L}_{recon}(\phi, \omega)$. Substituting $\mathcal{E}(\overrightarrow{\boldsymbol{x}_0}; \phi) = \mathcal{S}^T \overrightarrow{\boldsymbol{x}_0}$ and $\mathcal{D}(\overleftarrow{\boldsymbol{z}_0}; \omega) = \mathcal{S}\overleftarrow{\boldsymbol{z}_0}$, we have

$$
\begin{aligned}
\mathcal{L}_{recon}(\phi, \omega) &:= \mathbb{E}_{\overrightarrow{\boldsymbol{x}_0} \sim p}\left[\left\|\mathcal{D}(\mathcal{E}(\overrightarrow{\boldsymbol{x}_0}; \phi); \omega) - \overrightarrow{\boldsymbol{x}_0}\right\|_2^2\right] \\
&= \mathbb{E}_{\overrightarrow{\boldsymbol{x}_0} \sim p}\left[\left\|\mathcal{D}(\mathcal{S}^T \overrightarrow{\boldsymbol{x}_0}; \omega) - \overrightarrow{\boldsymbol{x}_0}\right\|_2^2\right] \\
&= \mathbb{E}_{\overrightarrow{\boldsymbol{x}_0} \sim p}\left[\left\|\mathcal{S}\mathcal{S}^T \overrightarrow{\boldsymbol{x}_0} - \overrightarrow{\boldsymbol{x}_0}\right\|_2^2\right]
\end{aligned}
$$

Using the fact that $\overrightarrow{\boldsymbol{x}_0}$ lives in a linear subspace, we arrive at

$$
\begin{aligned}
\mathcal{L}_{recon}(\phi, \omega) &= \mathbb{E}_{\overrightarrow{\boldsymbol{x}_0} \sim p}\left[\left\|\mathcal{S}\mathcal{S}^T \mathcal{S}\overrightarrow{\boldsymbol{z}_0} - \mathcal{S}\overrightarrow{\boldsymbol{z}_0}\right\|_2^2\right] \\
&\overset{(i)}{=} \mathbb{E}_{\overrightarrow{\boldsymbol{z}_0} \sim \mathcal{N}(\mathbf{0}, \boldsymbol{I}_k)}\left[\left\|\mathcal{S}\overrightarrow{\boldsymbol{z}_0} - \mathcal{S}\overrightarrow{\boldsymbol{z}_0}\right\|_2^2\right] = 0,
\end{aligned}
$$

where (i) is due to **Assumption 3.1**. Now, we analyze the distribution loss. Note that the KL-divergence between two Gaussian distributions with moments $(\mu_1, \sigma_1)$ and $(\mu_2, \sigma_2)$ is given by

$$
KL\left(\mathcal{N}(\mu_1, \sigma_1), \mathcal{N}(\mu_2, \sigma_2)\right) = \log\left(\frac{\sigma_2}{\sigma_1}\right) + \frac{\sigma_1^2 + (\mu_1 - \mu_2)^2}{2\sigma_2^2} - \frac{1}{2}.
$$

Since $\mathcal{E}(\boldsymbol{x}_0) = \mathcal{S}^T \boldsymbol{x}_0 = \mathcal{S}^T \mathcal{S}\boldsymbol{z}_0 = \boldsymbol{z}_0$, the distribution loss becomes:

$$
\mathcal{L}_{dist}(\phi) := KL\left(\mathcal{E}\sharp p, \mathcal{N}(\mathbf{0}, \boldsymbol{I}_k)\right) = KL\left(\mathcal{N}(\mathbf{0}, \boldsymbol{I}_k), \mathcal{N}(\mathbf{0}, \boldsymbol{I}_k)\right) = 0.
$$

$\square$

## B.3 Proof of Theorem 3.4

**Theorem B.3** (Generative Modeling using Diffusion in Latent Space)**.** *Suppose **Assumption 3.1** holds. Let the optimal solution of the latent diffusion model be*

$$
\boldsymbol{\theta}^* = \arg\min_{\boldsymbol{\theta}} \mathbb{E}_{\overrightarrow{\boldsymbol{z}_0}, \overrightarrow{\boldsymbol{\epsilon}}}\left[\left\|\tilde{\mu}_1\left(\overrightarrow{\boldsymbol{z}_1}(\overrightarrow{\boldsymbol{z}_0}, \overrightarrow{\boldsymbol{\epsilon}}), \overrightarrow{\boldsymbol{z}_0}\right) - \mu_\theta\left(\overrightarrow{\boldsymbol{z}_1}(\overrightarrow{\boldsymbol{z}_0}, \overrightarrow{\boldsymbol{\epsilon}})\right)\right\|^2\right].
$$

*For a fixed variance $\beta > 0$, if $\mu_\theta\left(\overrightarrow{\boldsymbol{z}_1}(\overrightarrow{\boldsymbol{z}_0}, \overrightarrow{\boldsymbol{\epsilon}})\right) := \boldsymbol{\theta}\overrightarrow{\boldsymbol{z}_1}(\overrightarrow{\boldsymbol{z}_0}, \overrightarrow{\boldsymbol{\epsilon}})$, then the closed-form solution is $\boldsymbol{\theta}^* = \sqrt{1-\beta}\boldsymbol{I}_k$, which after normalization by $\frac{1}{\sqrt{1-\beta}}$ and composition with the decoder $\mathcal{D}\left(\overleftarrow{\boldsymbol{z}_0}; \omega\right) := \mathcal{S}\overleftarrow{\boldsymbol{z}_0}$ recovers the true subspace of $p\left(\overrightarrow{\boldsymbol{x}_0}\right)$.*

*Proof.* In latent diffusion models, the training is performed in the latent space of a pre-trained VAE. If the VAE is chosen from **Proposition 3.3**, then the training objective becomes:

$$
\begin{aligned}
\min_{\boldsymbol{\theta}} &\mathbb{E}_{\overrightarrow{\boldsymbol{x}_0}, \overrightarrow{\boldsymbol{\epsilon}}}\left[\left\|\tilde{\mu}_1(\overrightarrow{\boldsymbol{z}_1}\left(\mathcal{E}(\overrightarrow{\boldsymbol{x}_0}), \overrightarrow{\boldsymbol{\epsilon}}\right), \mathcal{E}(\overrightarrow{\boldsymbol{x}_0})) - \mu_\theta\left(\overrightarrow{\boldsymbol{z}_1}\left(\mathcal{E}(\overrightarrow{\boldsymbol{x}_0}), \overrightarrow{\boldsymbol{\epsilon}}\right)\right)\right\|^2\right] \\
&= \mathbb{E}_{\overrightarrow{\boldsymbol{z}_0}, \overrightarrow{\boldsymbol{\epsilon}}}\left[\left\|\tilde{\mu}_1(\overrightarrow{\boldsymbol{z}_1}\left(\overrightarrow{\boldsymbol{z}_0}, \overrightarrow{\boldsymbol{\epsilon}}\right), \overrightarrow{\boldsymbol{z}_0}) - \mu_\theta\left(\overrightarrow{\boldsymbol{z}_1}\left(\overrightarrow{\boldsymbol{z}_0}, \overrightarrow{\boldsymbol{\epsilon}}\right)\right)\right\|^2\right] \\
&= \mathbb{E}_{\overrightarrow{\boldsymbol{z}_0}, \overrightarrow{\boldsymbol{\epsilon}}}\left[\left\|\overrightarrow{\boldsymbol{z}_0} - \mu_\theta\left(\overrightarrow{\boldsymbol{z}_1}\left(\overrightarrow{\boldsymbol{z}_0}, \overrightarrow{\boldsymbol{\epsilon}}\right)\right)\right\|^2\right] = \mathbb{E}_{\overrightarrow{\boldsymbol{z}_0}, \overrightarrow{\boldsymbol{\epsilon}}}\left[\left\|\overrightarrow{\boldsymbol{z}_0} - \boldsymbol{\theta}\overrightarrow{\boldsymbol{z}_1}\left(\overrightarrow{\boldsymbol{z}_0}, \overrightarrow{\boldsymbol{\epsilon}}\right)\right\|^2\right] \\
&= \mathbb{E}_{\overrightarrow{\boldsymbol{z}_0}, \overrightarrow{\boldsymbol{\epsilon}}}\left[\left\|\overrightarrow{\boldsymbol{z}_0} - \boldsymbol{\theta}\left(\overrightarrow{\boldsymbol{z}_0}\sqrt{1-\beta} + \sqrt{\beta}\overrightarrow{\boldsymbol{\epsilon}}\right)\right\|^2\right] \\
&= \mathbb{E}_{\substack{\overrightarrow{\boldsymbol{z}_0} \sim p \\ \overrightarrow{\boldsymbol{\epsilon}} \sim \mathcal{N}(\mathbf{0}, \boldsymbol{I}_k)}}\left[\sum_{i=1}^k \left(\overrightarrow{\boldsymbol{z}_{0,i}} - \boldsymbol{\theta}_i^T\left(\overrightarrow{\boldsymbol{z}_0}\sqrt{1-\beta} + \overrightarrow{\boldsymbol{\epsilon}}\sqrt{\beta}\right)\right)^2\right],
\end{aligned}
$$

where $\boldsymbol{\theta}_i^T$ denotes the $i^{th}$ row of matrix $\boldsymbol{\theta}$. The solution of this regression problem is given by[7]

$$
\begin{aligned}
\boldsymbol{\theta}_i^* &= \mathbb{E}_{\boldsymbol{x}_0,\boldsymbol{\epsilon}} \left[ \left( \boldsymbol{z}_0\sqrt{1-\beta} + \boldsymbol{\epsilon}\sqrt{\beta} \right) \left( \boldsymbol{z}_0\sqrt{1-\beta} + \boldsymbol{\epsilon}\sqrt{\beta} \right)^T \right]^{-1} \mathbb{E}_{\boldsymbol{x}_0,\boldsymbol{\epsilon}} \left[ \boldsymbol{z}_{0,i} \left( \boldsymbol{z}_0\sqrt{1-\beta} + \boldsymbol{\epsilon}\sqrt{\beta} \right) \right] \\
&= \mathbb{E}_{\boldsymbol{x}_0,\boldsymbol{\epsilon}} \left[ \left( \boldsymbol{z}_0\sqrt{1-\beta} + \boldsymbol{\epsilon}\sqrt{\beta} \right) \left( \boldsymbol{z}_0\sqrt{1-\beta} + \boldsymbol{\epsilon}\sqrt{\beta} \right)^T \right]^{-1} \mathbb{E}_{\boldsymbol{x}_0,\boldsymbol{\epsilon}} \left[ \boldsymbol{z}_{0,i} \left( \boldsymbol{z}_0\sqrt{1-\beta} + \boldsymbol{\epsilon}\sqrt{\beta} \right) \right] \\
&= \mathbb{E}_{\boldsymbol{x}_0,\boldsymbol{\epsilon}} \left[ \left( \mathcal{E}(\boldsymbol{x}_0)\sqrt{1-\beta} + \boldsymbol{\epsilon}\sqrt{\beta} \right) \left( \mathcal{E}(\boldsymbol{x}_0)\sqrt{1-\beta} + \boldsymbol{\epsilon}\sqrt{\beta} \right)^T \right]^{-1} \mathbb{E}_{\boldsymbol{x}_0,\boldsymbol{\epsilon}} \left[ \mathcal{E}(\boldsymbol{x}_0)_i \left( \mathcal{E}(\boldsymbol{x}_0)\sqrt{1-\beta} + \boldsymbol{\epsilon}\sqrt{\beta} \right) \right] \\
&= \mathbb{E}_{\boldsymbol{z}_0,\boldsymbol{\epsilon}} \left[ \left( \mathcal{E}(\mathcal{S}\boldsymbol{z}_0)\sqrt{1-\beta} + \boldsymbol{\epsilon}\sqrt{\beta} \right) \left( \mathcal{E}(\mathcal{S}\boldsymbol{z}_0)\sqrt{1-\beta} + \boldsymbol{\epsilon}\sqrt{\beta} \right)^T \right]^{-1} \mathbb{E}_{\boldsymbol{z}_0,\boldsymbol{\epsilon}} \left[ \mathcal{E}(\mathcal{S}\boldsymbol{z}_0)_i \left( \mathcal{E}(\mathcal{S}\boldsymbol{z}_0)\sqrt{1-\beta} + \boldsymbol{\epsilon}\sqrt{\beta} \right) \right] \\
&= \mathbb{E}_{\boldsymbol{z}_0,\boldsymbol{\epsilon}} \left[ \left( \mathcal{S}^T\mathcal{S}\boldsymbol{z}_0\sqrt{1-\beta} + \boldsymbol{\epsilon}\sqrt{\beta} \right) \left( \mathcal{S}^T\mathcal{S}\boldsymbol{z}_0\sqrt{1-\beta} + \boldsymbol{\epsilon}\sqrt{\beta} \right)^T \right]^{-1} \mathbb{E}_{\boldsymbol{z}_0,\boldsymbol{\epsilon}} \left[ (\mathcal{S}^T\mathcal{S}\boldsymbol{z}_0)_i \left( \mathcal{S}^T\mathcal{S}\boldsymbol{z}_0\sqrt{1-\beta} + \boldsymbol{\epsilon}\sqrt{\beta} \right) \right]
\end{aligned}
$$

Using **Assumption 3.1**, the above expression simplifies to

$$
\begin{aligned}
\boldsymbol{\theta}_i^* &= \mathbb{E}_{\boldsymbol{z}_0,\boldsymbol{\epsilon}} \left[ \left( \boldsymbol{z}_0\sqrt{1-\beta} + \boldsymbol{\epsilon}\sqrt{\beta} \right) \left( \boldsymbol{z}_0\sqrt{1-\beta} + \boldsymbol{\epsilon}\sqrt{\beta} \right)^T \right]^{-1} \mathbb{E}_{\boldsymbol{z}_0,\boldsymbol{\epsilon}} \left[ (\boldsymbol{z}_0)_i \left( \boldsymbol{z}_0\sqrt{1-\beta} + \boldsymbol{\epsilon}\sqrt{\beta} \right) \right] \\
&= \mathbb{E}_{\boldsymbol{z}_0,\boldsymbol{\epsilon}} \left[ (1-\beta)\boldsymbol{z}_0\boldsymbol{z}_0^T + \boldsymbol{z}_0\boldsymbol{\epsilon}^T\sqrt{\beta(1-\beta)} + \boldsymbol{\epsilon}\boldsymbol{z}_0^T\sqrt{\beta(1-\beta)} + \beta\boldsymbol{\epsilon}\boldsymbol{\epsilon}^T \right]^{-1} \mathbb{E}_{\boldsymbol{z}_0,\boldsymbol{\epsilon}} \left[ (\boldsymbol{z}_0)_i \left( \boldsymbol{z}_0\sqrt{1-\beta} + \boldsymbol{\epsilon}\sqrt{\beta} \right) \right] \\
&= \left[ \left( (1-\beta)\mathbb{E}_{\boldsymbol{z}_0,\boldsymbol{\epsilon}} \left[ \boldsymbol{z}_0\boldsymbol{z}_0^T \right] + \mathbb{E}_{\boldsymbol{z}_0,\boldsymbol{\epsilon}} \left[ \boldsymbol{z}_0\boldsymbol{\epsilon}^T \right]\sqrt{\beta(1-\beta)} + \mathbb{E}_{\boldsymbol{z}_0,\boldsymbol{\epsilon}} \left[ \boldsymbol{\epsilon}\boldsymbol{z}_0^T \right]\sqrt{\beta(1-\beta)} + \beta\mathbb{E}_{\boldsymbol{z}_0,\boldsymbol{\epsilon}} \left[ \boldsymbol{\epsilon}\boldsymbol{\epsilon}^T \right] \right) \right]^{-1} \\
&\qquad\qquad\qquad\qquad\qquad\qquad \times \mathbb{E}_{\boldsymbol{z}_0,\boldsymbol{\epsilon}} \left[ (\boldsymbol{z}_0)_i \left( \boldsymbol{z}_0\sqrt{1-\beta} + \boldsymbol{\epsilon}\sqrt{\beta} \right) \right] \\
&= \left[ \left( (1-\beta)\boldsymbol{I}_k + \mathbb{E}_{\boldsymbol{z}_0}\left[\boldsymbol{z}_0\right]\mathbb{E}_{\boldsymbol{\epsilon}}\left[\boldsymbol{\epsilon}\right]^T\sqrt{\beta(1-\beta)} + \mathbb{E}_{\boldsymbol{\epsilon}}\left[\boldsymbol{\epsilon}\right]\mathbb{E}_{\boldsymbol{z}_0}\left[\boldsymbol{z}_0\right]^T\sqrt{\beta(1-\beta)} + \beta\boldsymbol{I}_k \right) \right]^{-1} \\
&\qquad\qquad\qquad\qquad\qquad\qquad \times \mathbb{E}_{\boldsymbol{z}_0,\boldsymbol{\epsilon}} \left[ (\boldsymbol{z}_0)_i \left( \boldsymbol{z}_0\sqrt{1-\beta} + \boldsymbol{\epsilon}\sqrt{\beta} \right) \right],
\end{aligned}
$$

where the last step uses the fact that $\boldsymbol{z}_0$ and $\boldsymbol{\epsilon}$ are independent Gaussian random vectors with zero mean and unit covariance. Simplifying further, we arrive at

$$
\begin{aligned}
\boldsymbol{\theta}_i^* &= \left[ (1-\beta)\boldsymbol{I}_k + \beta\boldsymbol{I}_k \right]^{-1} \mathbb{E}_{\boldsymbol{z}_0,\boldsymbol{\epsilon}} \left[ (\boldsymbol{z}_0)_i \left( \boldsymbol{z}_0\sqrt{1-\beta} + \boldsymbol{\epsilon}\sqrt{\beta} \right) \right] \\
&= \mathbb{E}_{\boldsymbol{z}_0,\boldsymbol{\epsilon}} \left[ (\boldsymbol{z}_0)_i \left( \boldsymbol{z}_0\sqrt{1-\beta} + \boldsymbol{\epsilon}\sqrt{\beta} \right) \right] \\
&= \mathbb{E}_{\boldsymbol{z}_0} \left[ (\boldsymbol{z}_0)_i\boldsymbol{z}_0\sqrt{1-\beta} \right] + \mathbb{E}_{\boldsymbol{z}_0,\boldsymbol{\epsilon}} \left[ (\boldsymbol{z}_0)_i\boldsymbol{\epsilon}\sqrt{\beta} \right] \\
&= \mathbb{E}_{\boldsymbol{z}_0} \left[ (\boldsymbol{z}_0)_i\boldsymbol{z}_0\sqrt{1-\beta} \right] + \mathbb{E}_{\boldsymbol{z}_0} \left[ (\boldsymbol{z}_0)_i \right]\mathbb{E}_{\boldsymbol{\epsilon}}\left[\boldsymbol{\epsilon}\right]\sqrt{\beta}.
\end{aligned}
$$

The final step follows from independence of $\boldsymbol{z}_0$ and $\boldsymbol{\epsilon}$. Since $\boldsymbol{z}_0$ and $\boldsymbol{\epsilon}$ are also $\mathcal{N}\left(\boldsymbol{0}, \boldsymbol{I}_k\right)$, we get

$$
\boldsymbol{\theta}_i^* = \mathbb{E}_{\boldsymbol{z}_0} \left[ (\boldsymbol{z}_0)_i\boldsymbol{z}_0\sqrt{1-\beta} \right] = \left[ 0, \ldots, 0, \sqrt{1-\beta}, 0, \ldots, 0 \right]^T,
$$

where the $i^{th}$ coordinate is $\sqrt{1-\beta}$ and zero everywhere else. Therefore, stacking all the rows together, we get $\boldsymbol{\theta}^* = \sqrt{1-\beta}\boldsymbol{I}_k$, which after normalization by $1/\sqrt{1-\beta}$ gives the desired result.

Next, we show that $\boldsymbol{\theta}^*$ recovers the true subspace of $\overrightarrow{\boldsymbol{x}_0} \sim p\left(\overrightarrow{\boldsymbol{x}_0}\right)$. When composed with the decoder of VAE, the generator of the LDM gives $\overleftarrow{\boldsymbol{x}_0} = \mathcal{D}\left(\boldsymbol{\theta}^*\overleftarrow{\boldsymbol{z}_1}\right) = \mathcal{D}\left(\boldsymbol{I}_k\overleftarrow{\boldsymbol{z}_1}\right) = \mathcal{S}\overleftarrow{\boldsymbol{z}_1}$. Since $\overleftarrow{\boldsymbol{z}_1} \sim \mathcal{N}\left(\boldsymbol{0}, \boldsymbol{I}_k\right)$, this completes the statement of the theorem. $\qquad\square$

## B.4 Proof of Theorem 3.5

Recall that the the latent-space GML-DPS (6) algorithm (based on the pixel-space DPS algorithm [11]) has three key steps. In the first step, it uses the normalized closed-form solution obtained in

---

[7]For ease of notation, we drop the forward arrow in the rest of this proof.

**Theorem 3.4** to perform one step of *denoising* by the reverse SDE. In the second step, it runs one step of gradient descent to satisfy the *measurements* in the pixel space. Finally, it takes one step of gradient descent on the *goodness* objective, which acts as a regularizer to ensure that the reconstructed image lies on the data manifold.

This can be formalized as:

$$\overleftarrow{z'_0} = \theta^* \overleftarrow{z_1} - \eta \nabla_{\overleftarrow{z_1}} \left\| \mathcal{A}\mathcal{D}(\overleftarrow{z_0}(\overleftarrow{z_1})) - y \right\|_2^2; \tag{8}$$

$$\overleftarrow{z_0} = \arg\min_{\overleftarrow{z'_0}} \left\| \overleftarrow{z'_0} - \mathcal{E}(\mathcal{D}(\overleftarrow{z'_0})) \right\|_2^2, \tag{9}$$

In practice, solving (9) can be difficult, and can be approximated via gradient descent. In our analysis however, we analyze the exact system of equations above, as (9) has a closed-form solution in the linear setting.

**Theorem B.4** (Posterior Sampling using Goodness Modified Latent DPS). *Suppose **Assumptions 3.1** and **Assumption 3.2** hold. Denote by $\sigma = \{\sigma_j\}_{j=1}^k$ the singular values of $(\mathcal{A}\mathcal{S})^T(\mathcal{A}\mathcal{S})$, i.e., $(\mathcal{A}\mathcal{S})^T(\mathcal{A}\mathcal{S}) = U\Sigma U^T \coloneqq U D(\sigma) U^T, U \in \mathbb{R}^{k \times k}$, and let*

$$\theta^* = \arg\min_{\theta} \mathbb{E}_{\overrightarrow{z_0}, \overrightarrow{\epsilon}} \left[ \left\| \tilde{\mu}_1 \left( \overrightarrow{z_1}(\overrightarrow{z_0}, \overrightarrow{\epsilon}), \overrightarrow{z_0} \right) - \mu_\theta \left( \overrightarrow{z_1} \left( \overrightarrow{z_0}, \overrightarrow{\epsilon} \right) \right) \right\|_2^2 \right].$$

*Suppose $\overrightarrow{x_0} \sim p(\overrightarrow{x_0})$. Given measurements $y = \mathcal{A}\overrightarrow{x_0}$ and any fixed variance $\beta \in (0,1)$, then with the (unique) step size $\eta = (1/2)U D(\eta_i)U^T, \eta_i = \{\eta_i^j = 1/2\sigma_j\}_{j=1}^k$, the GML-DPS algorithm (6) samples from the true posterior $p(\overrightarrow{x_0}|y)$ and exactly recovers the groundtruth sample, i.e., $\overleftarrow{x_0} = \overrightarrow{x_0}$.*

*Proof.* We start with the measurement consistency update (8) and then show that the solution obtained from (8) is already a minimizer of (9). Therefore, we have

$$
\begin{aligned}
\overleftarrow{z'_0} &= \theta^* \overleftarrow{z_1} - \eta \nabla_{\overleftarrow{z_1}} \left\| \mathcal{A}\mathcal{D}(\overleftarrow{z_0}(\overleftarrow{z_1})) - y \right\|_2^2 \\
&= I_k \overleftarrow{z_1} - \eta \nabla_{\overleftarrow{z_1}} \left\| \mathcal{A}\mathcal{D}(I_k \overleftarrow{z_1}) - y \right\|_2^2 \\
&= \overleftarrow{z_1} - \eta \nabla_{\overleftarrow{z_1}} \left\| \mathcal{A}\mathcal{S}\overleftarrow{z_1}) - y \right\|_2^2 \\
&= \overleftarrow{z_1} - \eta \nabla_{\overleftarrow{z_1}} \left\| \mathcal{A}\mathcal{S}\overleftarrow{z_1}) - y \right\|_2^2 \\
&\overset{(i)}{=} \overleftarrow{z_1} - \eta \nabla_{\overleftarrow{z_1}} \left\| \mathcal{A}\mathcal{S}\overleftarrow{z_1} - y \right\|_2^2 \\
&= \overleftarrow{z_1} - 2\eta \mathcal{S}^T \mathcal{A}^T \left( \mathcal{A}\mathcal{S}\overleftarrow{z_1} - y \right) \\
&= \overleftarrow{z_1} - 2\eta \mathcal{S}^T \mathcal{A}^T \mathcal{A}\mathcal{S}\overleftarrow{z_1} + 2\eta \mathcal{S}^T \mathcal{A}^T y \\
&= \overleftarrow{z_1} - 2\eta \mathcal{S}^T \mathcal{A}^T \mathcal{A}\mathcal{S}\overleftarrow{z_1} + 2\eta \mathcal{S}^T \mathcal{A}^T \mathcal{A}\overrightarrow{x_0} \\
&= \overleftarrow{z_1} - 2\eta \mathcal{S}^T \mathcal{A}^T \mathcal{A}\mathcal{S}\overleftarrow{z_1} + 2\eta \mathcal{S}^T \mathcal{A}^T \mathcal{A}\mathcal{S}\overrightarrow{z_0},
\end{aligned}
$$

where (i) is due to **Assumption 3.1**. By **Assumption 3.2**, $(\mathcal{A}\mathcal{S})^T(\mathcal{A}\mathcal{S})$ is a positive definite matrix and can be written as $U\Sigma U^T$:

$$
\begin{aligned}
\overleftarrow{z'_0} &= \overleftarrow{z_1} - 2\eta U\Sigma U^T \overleftarrow{z_1} + 2\eta U\Sigma U^T \overrightarrow{z_0} \\
&= \overleftarrow{z_1} - 2U D(\eta_i)U^T U\Sigma U^T \overleftarrow{z_1} + 2U D(\eta_i)U^T U\Sigma U^T \overrightarrow{z_0} \\
&= \overleftarrow{z_1} - 2U D(\eta_i)\Sigma U^T \overleftarrow{z_1} + 2U D(\eta_i)\Sigma U^T \overrightarrow{z_0} \\
&= \overleftarrow{z_1} - 2U D(\eta_i)D(\sigma)U^T \overleftarrow{z_1} + 2U D(\eta_i)D(\sigma)U^T \overrightarrow{z_0} \\
&= \overleftarrow{z_1} - 2U D(\eta_i \odot \sigma)U^T \overleftarrow{z_1} + 2U D(\eta_i \odot \sigma)U^T \overrightarrow{z_0}.
\end{aligned}
$$

Since $\eta_j^i = 1/2\sigma_j$, the above expression further simplifies to

$$\overleftarrow{z'_0} = \overleftarrow{z_1} - UU^T \overleftarrow{z_1} + UU^T \overrightarrow{z_0} = \overrightarrow{z_0}.$$

Next, we show that $\overleftarrow{z'_0}$ is already a minimizer of (9). This is a direct consequence of the encoder-decoder architecture of the VAE: $\mathcal{E}(\mathcal{D}(\overleftarrow{z'_0})) = \mathcal{S}^T \mathcal{S}\overleftarrow{z'_0} = \overleftarrow{z'_0}$. Hence, $\left\| \overleftarrow{z'_0} - \mathcal{E}(\mathcal{D}(\overleftarrow{z'_0})) \right\|^2 = 0$, and

consequently $\overleftarrow{z_0} = \overleftarrow{z_0'} - \gamma \nabla_{\overleftarrow{z_0'}} \left|\left| \overleftarrow{z_0'} - \mathcal{E}(\mathcal{D}(\overleftarrow{z_0'})) \right|\right|^2 = \overrightarrow{z_0}$. Thus, the reconstructed sample becomes $\overleftarrow{x_0} = \mathcal{D}(\overleftarrow{z_0}) = \mathcal{S}\overrightarrow{z_0} = \overrightarrow{x_0}$.

Furthermore, as $\left|\left| \overleftarrow{z_0'} - \mathcal{E}(\mathcal{D}(\overleftarrow{z_0'})) \right|\right|^2 = 0$ for all $\overleftarrow{z_0'}$, it is evident that the goodness objective cannot rectify the error incurred in the measurement update (8). For this reason, GML-DPS algorithm (6) requires the exact step size to sample from the posterior. $\square$

Beyond the linear setting, we also refer to Table 5 for experiments supporting this result.

### B.5 Proof of Theorem 3.6

Different from GML-DPS, PSLD **Algorithm 2** replaces the goodness objective (6) with the gluing objective (7), which can be formalized as:

$$\overleftarrow{z_0'} = \boldsymbol{\theta}^* \overleftarrow{z_1} - \eta \nabla_{\overleftarrow{z_1}} \left\| \mathcal{A}\mathcal{D}(\overleftarrow{z_0}(\overleftarrow{z_1})) - \boldsymbol{y} \right\|_2^2 ; \tag{10}$$

$$\overleftarrow{z_0} = \arg\min_{\overleftarrow{z_0'}} \left|\left| \overleftarrow{z_0'} - \mathcal{E}(\mathcal{A}^T \mathcal{A} \overrightarrow{z_0} + (\boldsymbol{I}_d - \mathcal{A}^T \mathcal{A})\mathcal{D}(\overleftarrow{z_0'})) \right|\right|_2^2 . \tag{11}$$

We again remind that solving the minimization problem (11) is hard in general, and can be *approximated* by gradient descent as typically followed in practice [11]. However, in a linear model setting, (11) has a closed-form solution which we derive to prove exact recovery.

**Theorem B.5** (Posterior Sampling using Diffusion in Latent Space)**.** *Let **Assumptions 3.1** and **3.2** hold. Let $\sigma_j, \forall j = 1, \ldots, r$ denote the singular values of $(\mathcal{A}\mathcal{S})^T(\mathcal{A}\mathcal{S})$ and let*

$$\boldsymbol{\theta}^* = \arg\min_{\boldsymbol{\theta}} \mathbb{E}_{\overrightarrow{z_0}, \overrightarrow{\epsilon}} \left[ \left\| \tilde{\mu}_1\left(\overrightarrow{z_1}(\overrightarrow{z_0}, \overrightarrow{\epsilon}), \overrightarrow{z_0}\right) - \mu_\theta\left(\overrightarrow{z_1}\left(\overrightarrow{z_0}, \overrightarrow{\epsilon}\right)\right) \right\|^2 \right].$$

*Suppose $\overrightarrow{x_0} \sim p(\overrightarrow{x_0})$. Given measurements $\boldsymbol{y} = \mathcal{A}\overrightarrow{x_0}$, any fixed variance $\beta \in (0, 1)$, and any positive step sizes $\eta_i^j, j = 1, 2, \ldots, r$, the PSLD Algorithm 2 samples from the true posterior $p(\overrightarrow{x_0}|y)$ and exactly recovers the groundtruth sample, i.e., $\overleftarrow{x_0} = \overrightarrow{x_0}$.*

*Proof.* Following the proof in Appendix B.4, we have

$$\begin{aligned}
\overleftarrow{z_0'} &= \boldsymbol{\theta}^* \overleftarrow{z_1} - \eta \nabla_{\overleftarrow{z_1}} \left\| \mathcal{A}\mathcal{D}(\overleftarrow{z_0}(\overleftarrow{z_1})) - \boldsymbol{y} \right\|_2^2 \\
&= \boldsymbol{I}_k \overleftarrow{z_1} - \eta \nabla_{\overleftarrow{z_1}} \left\| \mathcal{A}\mathcal{D}(\overleftarrow{z_1}) - \boldsymbol{y} \right\|_2^2 \\
&= \overleftarrow{z_1} - \eta \nabla_{\overleftarrow{z_1}} \left\| \mathcal{A}\mathcal{S}\overleftarrow{z_1} - \boldsymbol{y} \right\|_2^2 \\
&= \overleftarrow{z_1} - 2\eta \mathcal{S}^T \mathcal{A}^T (\mathcal{A}\mathcal{S}\overleftarrow{z_1} - \boldsymbol{y}) \\
&= \overleftarrow{z_1} - 2\eta \mathcal{S}^T \mathcal{A}^T \mathcal{A}\mathcal{S}\overleftarrow{z_1} + 2\eta \mathcal{S}^T \mathcal{A}^T \mathcal{A}\mathcal{S}\overrightarrow{z_0} \\
&= \overleftarrow{z_1} - 2\eta \mathcal{S}^T \mathcal{A}^T \mathcal{A}\mathcal{S}\overleftarrow{z_1} + 2\eta \mathcal{S}^T \mathcal{A}^T \mathcal{A}\mathcal{S}\overrightarrow{z_0}.
\end{aligned}$$

We use the above expression to derive a closed-form solution to the minimization problem (11):

$$\begin{aligned}
\boldsymbol{0} &= \nabla_{\overleftarrow{z_0'}} \left|\left| \overleftarrow{z_0'} - \mathcal{S}^T(\mathcal{A}^T \mathcal{A}\mathcal{S}\overrightarrow{z_0} + (\boldsymbol{I}_d - \mathcal{A}^T \mathcal{A})\mathcal{S}\overleftarrow{z_0'}) \right|\right|_2^2 \\
&= \nabla_{\overleftarrow{z_0'}} \left|\left| \overleftarrow{z_0'} - \mathcal{S}^T \mathcal{A}^T \mathcal{A}\mathcal{S}\overrightarrow{z_0} - \mathcal{S}^T(\boldsymbol{I}_d - \mathcal{A}^T \mathcal{A})\mathcal{S}\overleftarrow{z_0'} \right|\right|_2^2 \\
&= \nabla_{\overleftarrow{z_0'}} \left|\left| \overleftarrow{z_0'} - \mathcal{S}^T \mathcal{A}^T \mathcal{A}\mathcal{S}\overrightarrow{z_0} - \mathcal{S}^T \mathcal{S}\overleftarrow{z_0'} - \mathcal{S}^T \mathcal{A}^T \mathcal{A}\mathcal{S}\overleftarrow{z_0'} \right|\right|_2^2 \\
&= \nabla_{\overleftarrow{z_0'}} \left|\left| \overleftarrow{z_0'} - \mathcal{S}^T \mathcal{A}^T \mathcal{A}\mathcal{S}\overrightarrow{z_0} - \mathcal{S}^T \mathcal{S}\overleftarrow{z_0'} + \mathcal{S}^T \mathcal{A}^T \mathcal{A}\mathcal{S}\overleftarrow{z_0'} \right|\right|_2^2 \\
&= 2\left(\boldsymbol{I}_k - \mathcal{S}^T \mathcal{S} + \mathcal{S}^T \mathcal{A}^T \mathcal{A}\mathcal{S}\right)\left(\overleftarrow{z_0'} - \mathcal{S}^T \mathcal{A}^T \mathcal{A}\mathcal{S}\overrightarrow{z_0} - \mathcal{S}^T \mathcal{S}\overleftarrow{z_0'} + \mathcal{S}^T \mathcal{A}^T \mathcal{A}\mathcal{S}\overleftarrow{z_0'}\right) \\
&= 2\mathcal{S}^T \mathcal{A}^T \mathcal{A}\mathcal{S}\left(\mathcal{S}^T \mathcal{A}^T \mathcal{A}\mathcal{S}\overleftarrow{z_0'} - \mathcal{S}^T \mathcal{A}^T \mathcal{A}\mathcal{S}\overrightarrow{z_0}\right),
\end{aligned}$$

where the last step is due to **Assumption 3.1**. Thus, we have

$$\overleftarrow{z_0} = \arg\min_{\overleftarrow{z_0'}} \left|\left| \overleftarrow{z_0'} - \mathcal{E}(\mathcal{A}^T \mathcal{A}\overrightarrow{z_0} + (\boldsymbol{I}_d - \mathcal{A}^T \mathcal{A})\mathcal{D}(\overleftarrow{z_0'})) \right|\right|_2^2 = \overrightarrow{z_0},$$

which produces $\overleftarrow{\boldsymbol{x}}_0 = \mathcal{D}(\overleftarrow{\boldsymbol{z}}_0) = \mathcal{D}(\overrightarrow{\boldsymbol{z}}_0) = \mathcal{S}\overrightarrow{\boldsymbol{z}}_0 = \overrightarrow{\boldsymbol{x}}_0.$ □

It is worth highlighting that PSLD exactly recovers the groundtruth sample irrespective of the choice of the step size $\eta$, whereas GML-DPS requires the step size to be exactly $\boldsymbol{\eta} = (1/2)\boldsymbol{U}\boldsymbol{D}(\boldsymbol{\eta}_i)\boldsymbol{U}^T$.

## C   Additional Experiments

### C.1   Implementation Details

For inpainting tasks, we note that the PSLD sampler generates missing parts (by design of our gluing objective) that are consistent with the known portions of the image, i.e., $\overleftarrow{\boldsymbol{x}}_0 = \mathcal{A}^T\mathcal{A}\overrightarrow{\boldsymbol{x}}_0 + (\boldsymbol{I}_d - \mathcal{A}^T\mathcal{A})\mathcal{D}(\overleftarrow{\boldsymbol{z}}_0)$. This is different from the DPS sampler, which generates the whole image which may not match the observations exactly. In other words, in the last of step of our algorithm, the observations are glued onto the corresponding parts of the generated image, leaving the unmasked portions untouched [54]. This sometimes creates edge effects which are then removed by post-processing the glued image through the encoder and decoder of the SD model, i.e. running one last step of our algorithm. Figure 2 illustrates that gluing the observations in commercial services still leads to visually inconsistent results (e.g. head in top row) unlike our method.

For all other tasks, such as motion deblur, Gaussian deblur, and super-resolution, this last step is not needed, as there is no box inpainting, i.e., $\overleftarrow{\boldsymbol{x}}_0 = \mathcal{D}(\overleftarrow{\boldsymbol{z}}_0)$. Furthermore, we use the same measurement operator $\mathcal{A}$ and its transpose $\mathcal{A}^T$ as provided by the DPS code repository[8]. However, since Stable Diffusion v1.5 generates images of size $512 \times 512$ resolution and DPS operates at $256 \times 256$, we adjust the size of the kernels used in PSLD to ensure that both the methods use the same amount of information while sampling from the posterior. During evaluation, we downsample PSLD generated images from $512 \times 512$ to $256 \times 256$ to compare with DPS at the same resolution.

**PSLD (Stable Diffusion-V1.5 ):** We run **Algorithm 2** with Stable Diffusion version 1.5 as the foundation model[9]. We use a fixed $\eta = 1$ and $\gamma = 0.1$. Since we study posterior sampling of images without conditioning on *text* inputs, we pass an empty string to the Stable Diffusion foundation model, which accepts texts as an input argument. For better performance, we recommend using the latest pretrained weights.

**PSLD (LDM-VQ-4 ):** This is the same sampling algorithm as before but with a different latent diffusion model, LDM-VQ-4[10] , which contains pretrained weights for FFHQ 256[11] and large-scale text-to-image generative model[12]. We keep the hyperparameters same ($\eta = 1$ and $\gamma = 0.1$). For each task, we provide hyper-parameter details in our codebase[13]. Although we have tested our framework with these two latent-diffusion-models, one may experiment with other latent-diffusion-models available in the same repository.

**DPS:** We use the original source code provided by the authors[14].

**OOD images are sourced online:**

1. Figure 1: the original images are generated by Stable Diffusion v-2.1[15].
2. Figure 2 first row: Walking example from the web.
3. Figure 2 second row, Obama-Biden image from the web.
4. Figure 2 third row, Fisherman from ImageNet 256 [17].
5. Figure 4 first row: Racoon image from the web.
6. Figure 4 second row: Fisherman from ImageNet 256 [17].
7. Figure 15: Celebrity face from the web.

---

[8]https://github.com/DPS2022/diffusion-posterior-sampling/blob/main/guided_diffusion/measurements.py
[9]https://huggingface.co/runwayml/stable-diffusion-v1-5
[10]https://github.com/CompVis/latent-diffusion
[11]https://ommer-lab.com/files/latent-diffusion/ffhq.zip
[12]https://ommer-lab.com/files/latent-diffusion/nitro/txt2img-f8-large/model.ckpt
[13]https://github.com/LituRout/PSLD
[14]https://github.com/DPS2022/diffusion-posterior-sampling
[15]https://huggingface.co/spaces/stabilityai/stable-diffusion

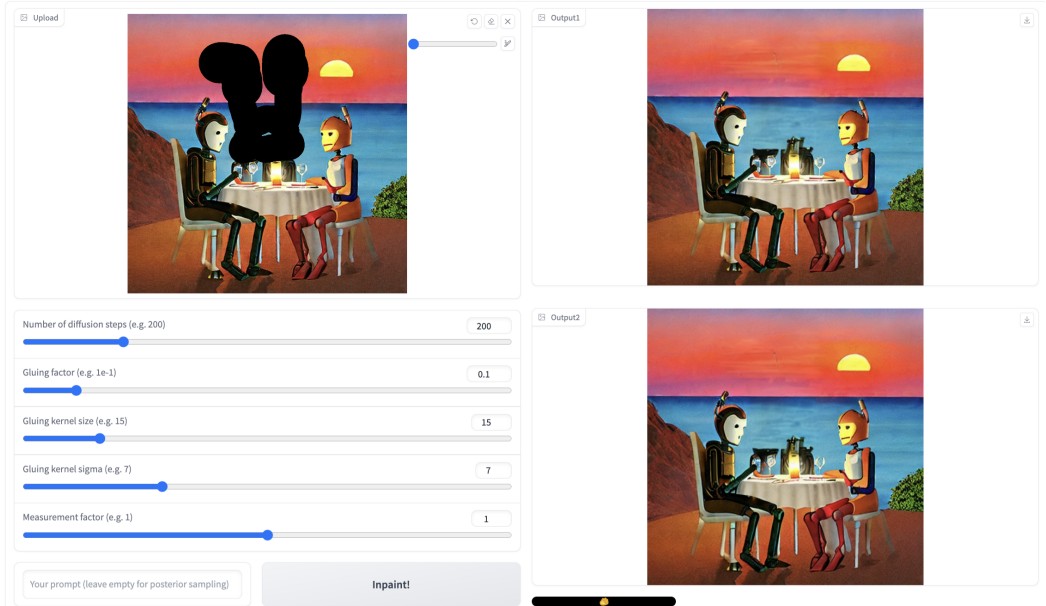

Figure 6: Results from the web application of our PSLD algorithm, $512 \times 512$. The original image (1) is generated by Stable Diffusion v-2.1 with the prompt,"*A dinner date between a robot couple during sunset*".

## C.2 Additional Experimental Evaluation

Here, we provide additional results to support our theoretical claims on various inverse problems.

Figures 6, 7, 8, and 9 show the inpainting results of user defined masks obtained from our PSLD inpainting web demo. Note that the foundation model used in this demo is a generic model. For better performance on specific images, we recommend finetuning the foundation model on this class and then running posterior sampling using our web demo: `https://huggingface.co/spaces/PSLD/PSLD`.

Figure 10 and 11 illustrate **super-resolution** ($4\times$) of in-distribution samples from the validation set of FFHQ 256. Observe that the samples generated by DPS are far from the groundtruth sample. On the other hand, the samples generated by PSLD closely capture the perceptual quality of the groundtruth sample. In other words, one may identify (b) and (c) as images of two different individuals, whereas (b) and (d) of the same individual. We attribute this *photorealism* of our method to the power of Stable Diffusion foundation model and the ability to use the knowledge of the VAE encoder-decoder in the gluing objective.

In addition, we test on out-of-distribution samples from ImageNet [17] validation set. Figure 12 and Figure 13 show the results in **motion deblur** and **Gaussian deblur**, respectively. By leveraging the foundation model Stable Diffusion v1.5, our PSLD method clearly outperforms DPS [11] in the general domain. Further, Figures 14, 15, and 16 show reconstruction of general domain samples for **random inpainting**, **super-resolution**, and **destriping** tasks, respectively. In all these tasks, the samples generated by PSLD are closer to the groundtruth sample than the ones generated by DPS. Figure 17 shows the results on image colorization. Table 5 and Table 6 show the quantitative results. Table 7 draws a comparison between the latent-DPS and PSLD algorithms, and shows that the PSLD objective enhances the reconstruction performance.

In Table 8, we compare the runtime and NFEs of PSLD with prior works. PSLD-SD (trained on LAION-5B) takes 776 s to generate 512x512 images. To compare with other methods that generate 256x256 images, we divide our runtime by 4. All the other methods use diffusion models trained on FFHQ and produce 256x256 images.

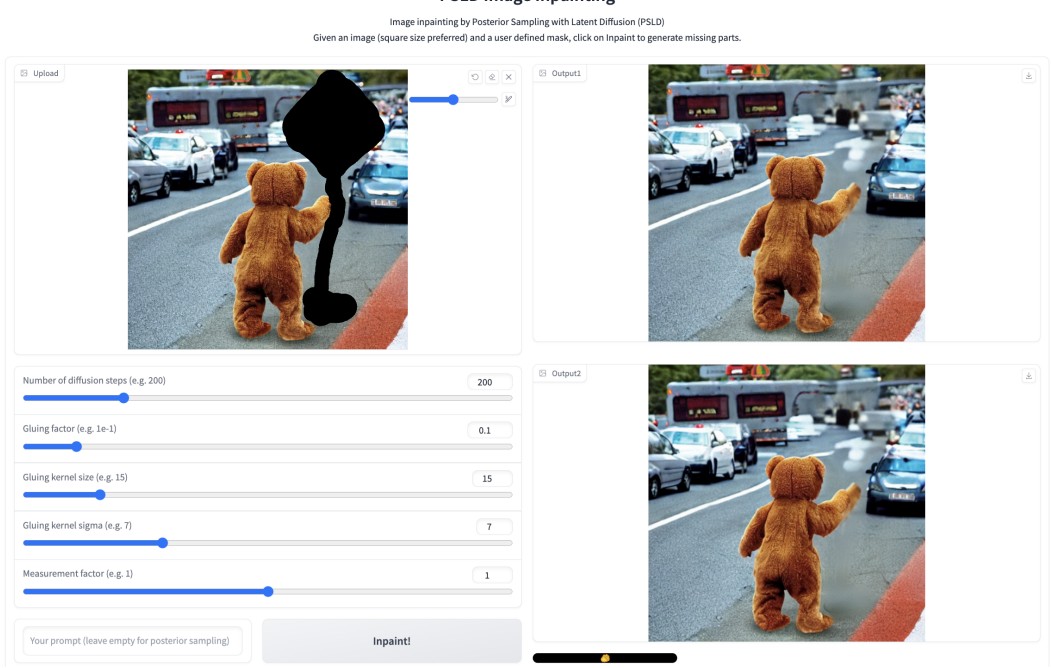

Figure 7: Results from the web application of our PSLD algorithm, $512 \times 512$. The original image (1) is generated by Stable Diffusion v-2.1 with the prompt,"*A panda wearing a spiderman costume*".

Figure 8: Results from the web application of our PSLD algorithm, $512 \times 512$. The original image (1) is generated by Stable Diffusion v-2.1 with the prompt,"*A teddy bear showing stop sign at the traffic*".

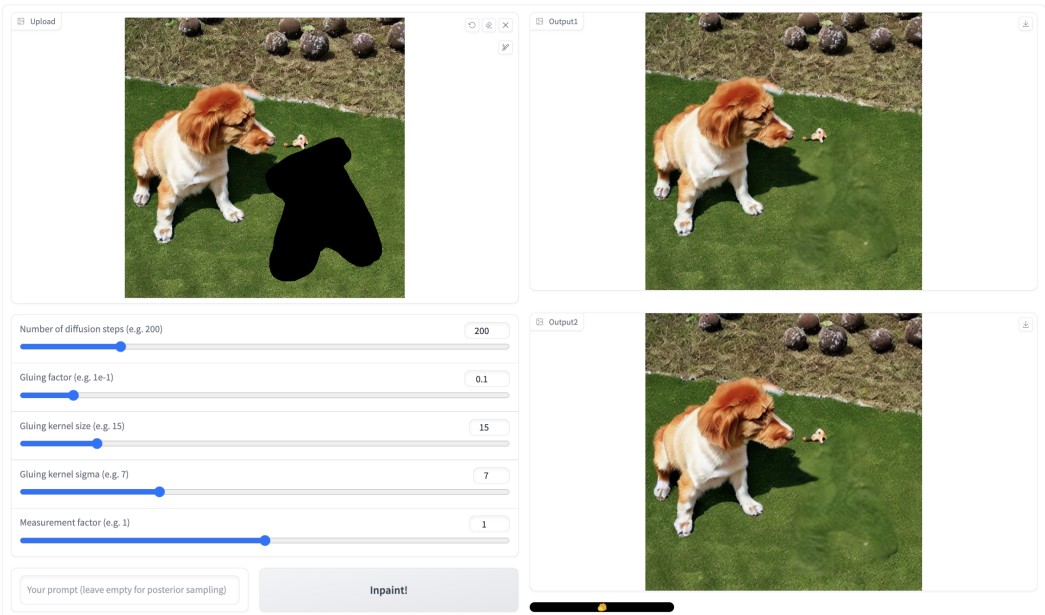

Figure 9: Results from the web application of our PSLD algorithm, $512 \times 512$. The original image (1) is generated by Stable Diffusion v-2.1 with the prompt,"*A cute dog playing with a toy teddy bear on the lawn*".

Table 5: Quantitative random inpainting results on FFHQ 256 validation set [25, 11]. We use Stable Diffusion (v1.5) trained on LAION.

| | Inpaint (random) | | SR ($4\times$) | | Gaussian Deblur | |
|---|---|---|---|---|---|---|
| Method | PSNR ($\uparrow$) | SSIM ($\uparrow$) | PSNR ($\uparrow$) | SSIM ($\uparrow$) | PSNR ($\uparrow$) | SSIM ($\uparrow$) |
| PSLD (Ours) | **33.71** | **0.943** | **30.73** | **0.867** | **30.10** | **0.843** |
| GML-DPS (Ours) | 29.49 | 0.844 | 29.77 | 0.860 | 29.21 | 0.820 |
| DPS [11] | 25.23 | **0.851** | 25.67 | 0.852 | 24.25 | 0.811 |
| DDRM [26] | 9.19 | 0.319 | 25.36 | 0.835 | 23.36 | 0.767 |
| MCG [13] | 21.57 | 0.751 | 20.05 | 0.559 | 6.72 | 0.051 |
| PnP-ADMM [6] | 8.41 | 0.325 | 26.55 | 0.865 | 24.93 | 0.812 |
| Score-SDE [50] | 13.52 | 0.437 | 17.62 | 0.617 | 7.12 | 0.109 |
| ADMM-TV | 22.03 | 0.784 | 23.86 | 0.803 | 22.37 | 0.801 |

## D   Additional Discussion

**Curse of ambient dimension:** DPS [11] suffers from the curse of ambient dimension because in this method, gradients are computed in the pixel space with dimension $d$. However, latent-based methods such as PSLD compute gradients in the latent dimension $k$, and hence the computation is more efficient. Furthermore, applying the chain rule on VAE and running diffusion in the latent space is less expensive than running diffusion in pixel space directly. In practice, the computational complexity of Stable Diffusion model ($\sim 4GB$) is higher (roughly 6 times) than the computational complexity of the encoder-decoder model ($\sim 700MB$). Therefore, applying the chain rule in the encoder-decoder and running diffusion in the latent space is less expensive than applying diffusion models in the pixel space directly.

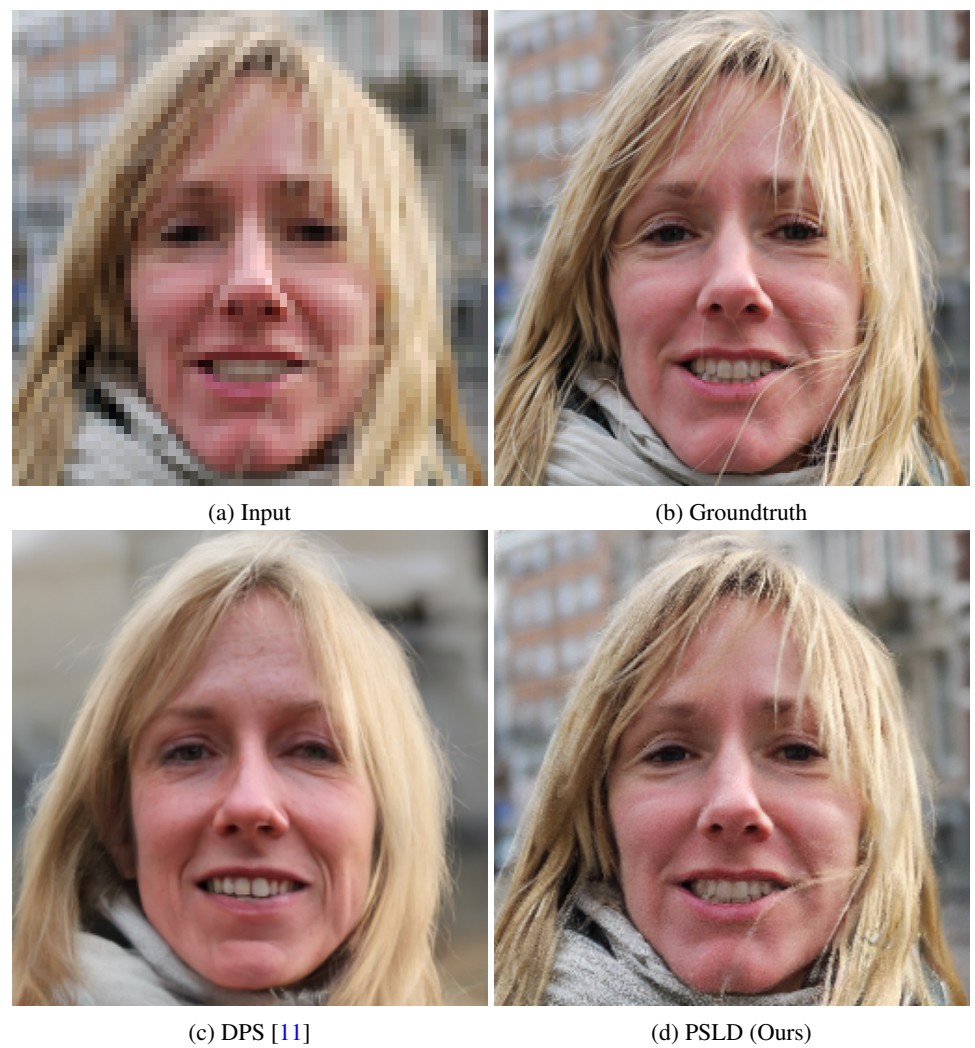

(a) Input              (b) Groundtruth

(c) DPS [11]              (d) PSLD (Ours)

Figure 10: Super-resolution results on images from FFHQ 256 [25, 11] (in distribution).

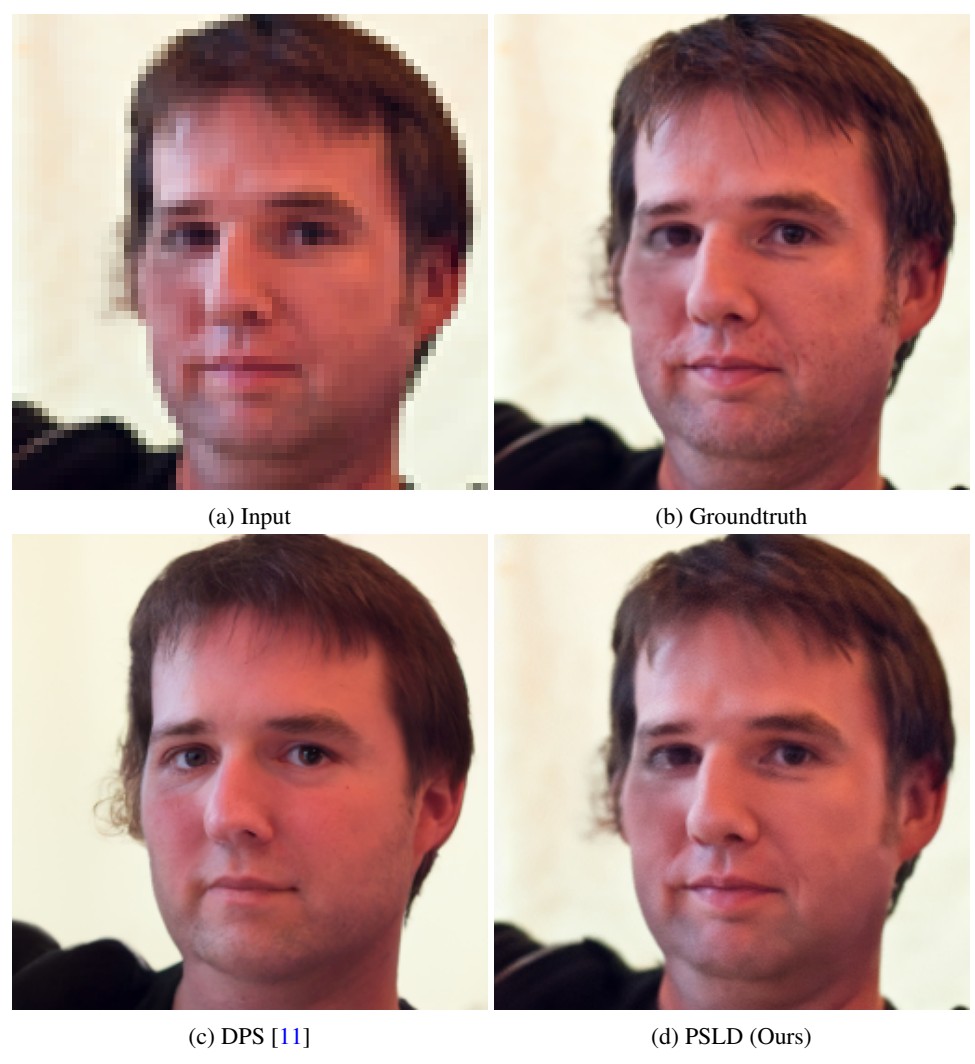

(a) Input

(b) Groundtruth

(c) DPS [11]

(d) PSLD (Ours)

Figure 11: Super-resolution results on FFHQ 256 [25, 11] (in distribution).

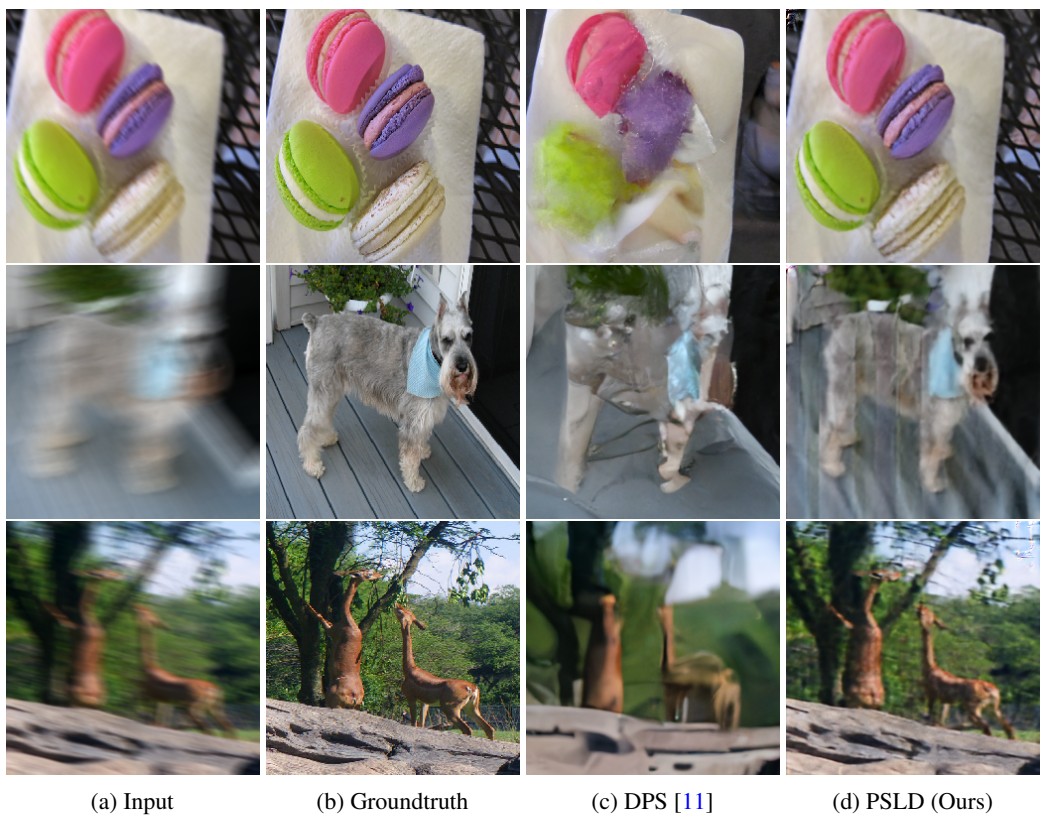

| (a) Input | (b) Groundtruth | (c) DPS [11] | (d) PSLD (Ours) |

Figure 12: Motion deblur results on ImageNet 256 [17] (out-of-distribution).

Table 6: Additional quantitative results on FFHQ 256 validation set [25, 11].

| | SR (4×) | | Gaussian Deblur | |
|---|---|---|---|---|
| Method | FID (↓) | LPIPS (↓) | FID (↓) | LPIPS (↓) |
| PSLD (Ours) | **34.28** | **0.201** | **41.53** | **0.221** |
| DPS [11] | 39.35 | 0.214 | 44.05 | 0.257 |
| DDRM [26] | 62.15 | 0.294 | 74.92 | 0.332 |
| MCG [13] | 87.64 | 0.520 | 101.2 | 0.340 |
| PnP-ADMM [6] | 66.52 | 0.353 | 90.42 | 0.441 |
| Score-SDE [50] | 96.72 | 0.563 | 109.0 | 0.403 |
| ADMM-TV | 110.6 | 0.428 | 186.7 | 0.507 |

| | SR (4×) | | Gaussian Deblur | |
|---|---|---|---|---|
| Method | PSNR (↑) | SSIM (↑) | PSNR (↑) | SSIM (↑) |
| PSLD (Ours) | **30.73** | **0.867** | **30.10** | **0.843** |
| GML-DPS (Ours) | 29.77 | 0.860 | 29.21 | 0.820 |
| DMPS [34] | 27.63 | - | 25.41 | - |
| DPS [11] | 25.67 | 0.852 | 24.25 | 0.811 |
| DDRM [26] | 25.36 | 0.835 | 23.36 | 0.767 |
| MCG [13] | 20.05 | 0.559 | 6.72 | 0.051 |
| PnP-ADMM [6] | 26.55 | 0.865 | 24.93 | 0.812 |
| Score-SDE [50] | 17.62 | 0.617 | 7.12 | 0.109 |
| ADMM-TV | 23.86 | 0.803 | 22.37 | 0.801 |

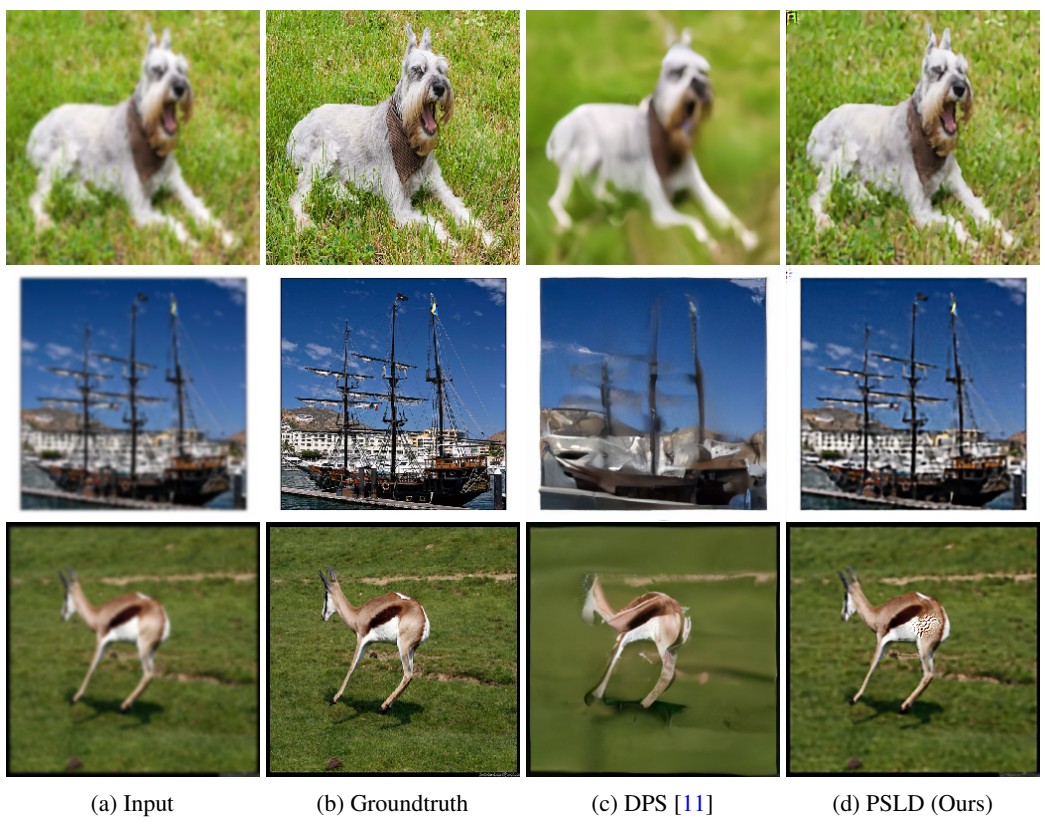

| (a) Input | (b) Groundtruth | (c) DPS [11] | (d) PSLD (Ours) |

Figure 13: Gaussian deblur results on ImageNet 256 [17] (out-of-distribution).

Table 7: Latent-DPS and PSLD methods evaluated on FFHQ 256 validation set [25, 11]. We use the *latent diffusion* (LDM-VQ-4) trained on FFHQ 256. Latent-DPS is a special case of PSLD algorithm when $\gamma = 0$.

| | Inpaint (box) | | |
|---|---|---|---|
| Method | PSNR (↑) | SSIM (↑) | LPIPS (↓) |
| PSLD | **24.22** | **0.819** | **0.158** |
| latent-DPS | 17.58 | 0.780 | 0.21 |

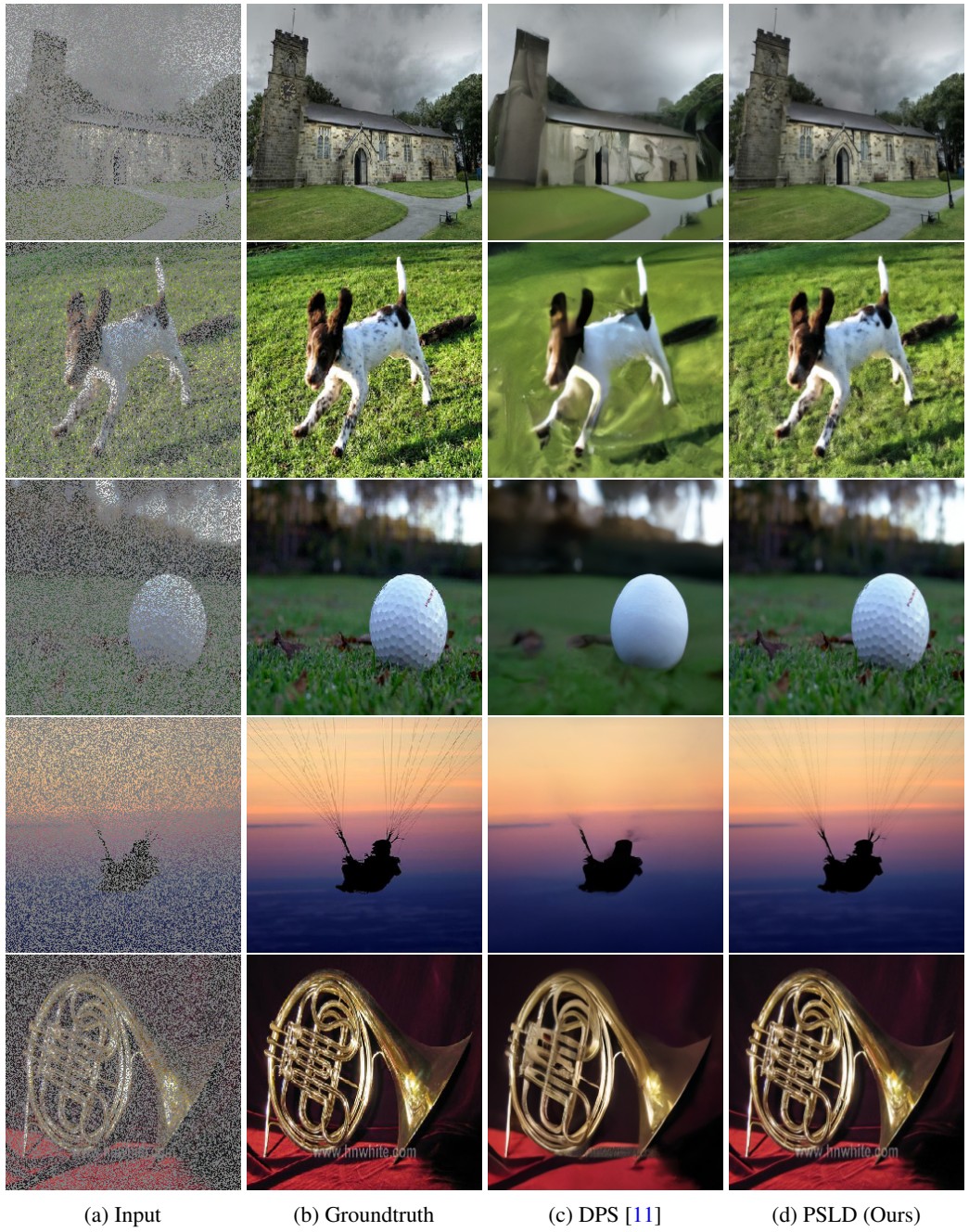

(a) Input         (b) Groundtruth         (c) DPS [11]         (d) PSLD (Ours)

Figure 14: Random inpainting results on ImageNet 256 [17] (out-of-distribution).

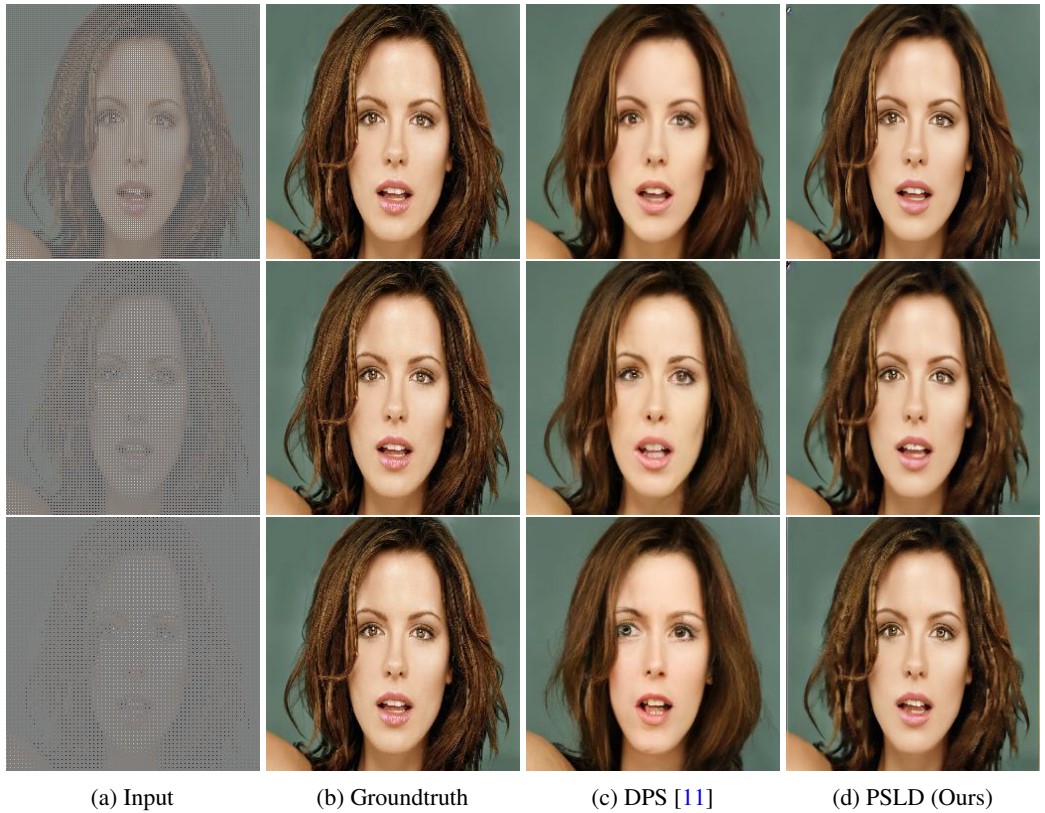

|            |                |         |              |
|------------|----------------|---------|--------------|
| (a) Input  | (b) Groundtruth | (c) DPS [11] | (d) PSLD (Ours) |

Figure 15: Super-resolution (using nearest neighbor kernel from [32]) results on out-of-distribution samples from the web, $256 \times 256$ (see Table 1 for LPIPS of these images).

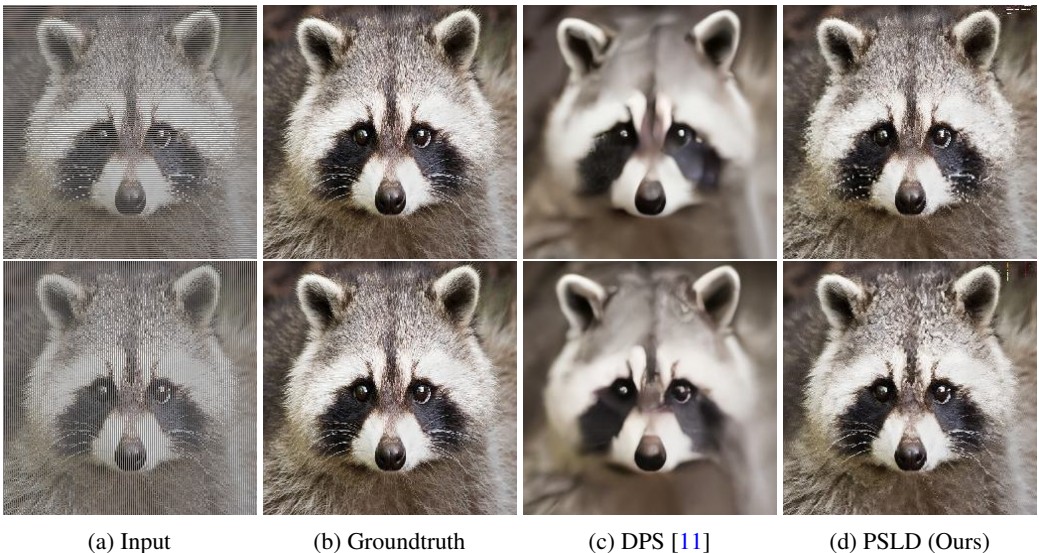

|            |                |         |              |
|------------|----------------|---------|--------------|
| (a) Input  | (b) Groundtruth | (c) DPS [11] | (d) PSLD (Ours) |

Figure 16: Destriping results on out-of-distribution samples from the web, $256 \times 256$. (**Top row**) Horizontal destriping: LPIPS of PSLD=0.244 and DPS [11]=0.613. (**Bottom row**) Vertical destriping: LPIPS of PSLD=0.255, DPS [11]=0.597.

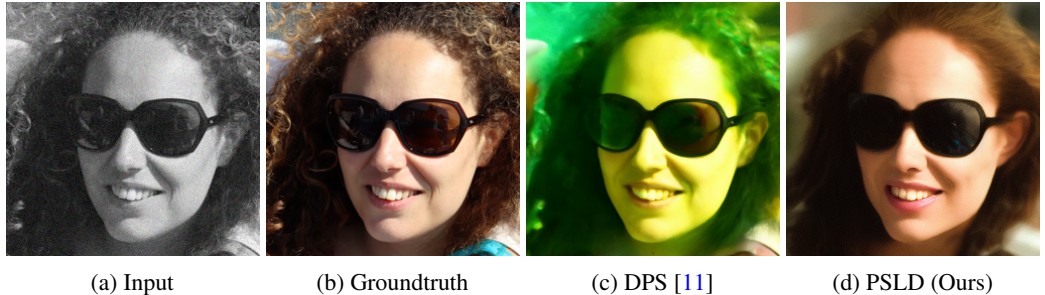

| (a) Input | (b) Groundtruth | (c) DPS [11] | (d) PSLD (Ours) |

Figure 17: Additional colorization results on images from FFHQ 256 [25, 11]. PSLD generates photo-realistic color, whereas DPS [11] generates overly saturated images.

Table 8: Runtime (top) and NFEs (bottom) of different posterior sampling algorithms. Runtimes are computed for the super-resolution task.

| Method | Runtime (s) |
|---|---|
| PSLD-LDM | 187.00 |
| PSLD-LDM (LAION-400M) | 190.00 |
| PSLD-SD (LAION-5B) | 194.25 |
| DMPS [34] | 67.02 |
| DPS [11] | 180.00 |
| DDNM+ [55] | 18.5 |
| DDRM [26] | 2.15 |
| MCG [13] | 193.71 |

| Method | NFEs |
|---|---|
| PSLD (Ours) | 100 to 1000 |
| DPS [11] | 1000 |
| DDRM [26] | 20 |
| RED [40] | 500 |
| $\Pi$GDM [46] | 20 to 100 |
| Palette [43] | 1000 |
| Regression | 1 |
| SNIPS [28] | 1000 |

