# OpenReview forum: "Solving Linear Inverse Problems Provably via Posterior Sampling with Latent Diffusion Models"
_NeurIPS.cc/2023/Conference — NeurIPS 2023 poster_

### Official Review · Reviewer_xKok · 2023-07-04

**Soundness:** 3 good
**Presentation:** 3 good
**Contribution:** 3 good
**Rating:** 6
**Confidence:** 4

**Summary:**

This paper proposes a method to deal with linear inverse problems using pre-trained latent diffusion models (LDM) as a prior. The main idea is to extend the original diffusion posterior sampling (DPS) to the case of LDM by approximating the gradient term of the intractable likelihood.  Two approximations (GLM-DPS), and (PSLD) are proposed for such approximation, and it turns out that PSLD achieves better performances. This paper also theoretically analyzes the DPS and PSLD in a toy linear setting. Experiment results show superior performances compared with the original DPS in a variety of tasks, such as random/block inpainting, denoising, super-resolution.

**Strengths:**

1. It is the first work, to my best knowledge, that LDM is used to address the linear inverse problems.

2. Compared to the original DPS, the proposed PSLD achieves apparently better performances in most cases.

3. Some theoretical analysis is also provided, although it is based on some toy linear model.

**Weaknesses:**

1.  There is a lack of analysis of the complexity, or running time, of the proposed PSLD method. Adding details of the running time of PSLD and other comparison methods is suggested.

2. Part 3 contains some theoretical analysis using a simple toy linear model. For example, Theorem 3.4 states that DPS exactly recovers the groundtruth sample. It is known that DPS uses a Laplace approximation to approximate the gradient term of the likelihood. As a result, how can it exactly recover the posterior samples with the crude approximation? This is also the case with Theorem 3.7 and Theorem 3.8.  To what extent can the results of Theorem 3.4, 3.7, and 3.8 be generalized to the real diffusion models?

3. In the experiments part, while PSLD shows superior performances over the original DPS, there is a lack of comparison with other latest diffusion-model-based algorithms that were available long before NeurIPS submission, e.g.,

[A] Wang, Yinhuai, Jiwen Yu, and Jian Zhang. "Zero-shot image restoration using denoising diffusion null-space model." arXiv preprint arXiv:2212.00490 (2022).

[B] Meng, Xiangming, and Yoshiyuki Kabashima. "Diffusion model based posterior sampling for noisy linear inverse problems." arXiv preprint arXiv:2211.12343 (2022).

[C] Song, Jiaming, Arash Vahdat, Morteza Mardani, and Jan Kautz. "Pseudoinverse-guided diffusion models for inverse problems." In International Conference on Learning Representations. 2022.

4. Most previous image restoration methods using diffusion models also consider colorization, can the authors also add some results in the case of colorization? Besides, only Table 2 and Table 5 show quantitative comparisons with other methods but only for inpainting. It is suggested to add quantitative comparisons with other methods in other tasks such as super-resolution, denoising, colorization, etc.

**Questions:**

Three additional questions:

1. How do you add the additive Gaussian noise with standard deviation \sigma_y?


2. Can I understand that the improved performance of PSLD over DPS comes solely from the improvement of diffusion models in the latent space over that in the pixel space? In other words, if DPS is implemented with more powerful diffusion models, is it possible to outperform PSLD? In this sense, how to ensure a fair comparison between PSLD and DPS?

3. Are the results of PSLD sensitive to the choice of hyper-parameters? Compared to DPS which has only one scaling hyperparameter, there are two scaling hyperparameters, namely, \eta and \gamma. It is suggested to evaluate the sensitivity of these hyperparameters.

**Limitations:**

Please refer to the weakness and questions parts.

---

> ### Author Rebuttal · Authors · 2023-08-09
>
> ## Response to Reviewer xKok
>
> Dear Reviewer xKok,
>
> Thank you for the review and for pointing out that **our study achieves state-of-the-art performance** in addressing inverse problems with latent diffusion models, and **unleashes the capacity of large-scale pre-trained LDMs** for sample recovery.
>
>
> Below, we provide answers to your remaining comments and questions.
>
> (1) **NFEs of different algorithms.**
> Since the runtime of diffusion based posterior samplers depends heavily on the number of Neural Function Evaluations (NFEs), we add comparison of different algorithms in terms NFEs.
>
> |    |PSLD (ours)|DPS |DDRM|RED [2]|$\Pi$GDM [1]|Palette [3]|Regression|SNIPS [4]|
> |:---|:----------|:--------|:--|:---|:--|:------|:---------|:----|
> |NFEs|100 to 1000|1000|20 |500|20 to 100|1000   |1         |1000 |
>
> The computational complexity of our method PSLD is similar to that of DPS. For a relatively smaller number of diffusion steps (say 100), PSLD significantly outperforms DPS. The DPS generated samples do not look like realistic images for fewer diffusion steps [1].
>
>
> **Runtime of different algorithms for Super-resolution task.**
>
> |Method        | Runtime (s)|
> |:-------------|:-----------|
> |DMPS [6]|67.02  |
> |DPS     |180.00 |
> |DDNM+[5]|18.5   |
> |DDRM    |2.15   |
> |MCG     |193.71 |
> |PSLD-LDM|187.00 |
> |PSLD-LDM (LAION-400M)^|190.00 |
> |PSLD-SD (LAION-5B)* |194.25  |
>
>
> *PSLD-SD (trained on LAION-5B) takes 776 s to generate 512x512 images. To compare with other methods, we divide its runtime by 4. ^PSLD-LDM (LAION-400M) uses a diffusion model trained on LAION-400M dataset. All the other methods use diffusion models trained on FFHQ and produce 256x256 images.
>
>
> (2) **Exact recovery is not possible by DPS.**
> Although DPS uses Laplace approximation for the gradient of the likelihood, this approximation is not necessary in a recoverable linear model setting as it can be exactly solved. The same argument also holds for Theorem 3.7 and Theorem 3.8.
>
> (3) **PSLD is better than DPS, but no comparison with other latest diffusion-model-based algorithms available before NeurIPS deadline.**
> Thanks for bringing these recent studies to our attention. Unfortunately, none of the suggested papers are **latent diffusion-model-based algorithms**. Nevertheless, we have compared with these methods, please see our answer to (7) and the attached PDF.
>
>
> (4) **How do you add Gaussian noise with standard deviation σ.**
> We sample $\hat{n} \sim \mathcal{N}(0,I)$ and then scale this sample by $\sigma$, i.e., $n = \sigma * \hat{n}~$ to generate the IID Gaussian noise $n \sim \mathcal{N}(0,\sigma I).$
>
> (5) **If DPS is implemented with a powerful diffusion model, can it outperform PSLD?**
> Our goal is to develop a framework to leverage the power of pre-trained latent diffusion models, such as Stable Diffusion. Our focus is not to maximize the margin that we beat the previous state-of-the-art DPS, but to unlock the potential of large pre-trained LDMs. Needless to say, the more powerful the prior the better it is for posterior sampling (e.g., classical priors to GANs and GANs to Diffusion). But previous posterior sampling algorithms (including DPS, DMPS and DDRM) only apply to pixel-space diffusion models, which leaves a gap to leverage the power of pre-trained LDMs. In this paper, we bridge this gap in literature as pointed out by Reviewer vbWT.
>
> (6) **Robustness of hyper-parameters.**
> The values of {$\gamma = 0.1$ and $\eta = 1$} are fixed for most of the practical tasks as mentioned in the appendix, justifying their robustness.
>
> (7) **Additional quantitative Results.**
> As suggested by the reviewer, we have added the following new results:
>
>
>
> | Method | SR(4X) |            | Gaussian Deblur |             |
> |:----------|:--------|:--------|:--------------------|:--------|
> |               |PSNR  |SSIM|PSNR|SSIM
> |PSLD (Ours)|30.73|0.867|30.10|0.843
> |GML-DPS (Ours)|29.77|0.860|29.21|0.820
> |DMPS[6]|27.63|-|25.41|-
> |DPS|25.67|0.852|24.25|0.811
> |DDRM|25.36|0.835|23.36|0.0.767
> |MCG|20.05|0.559|6.72|0.051
> |PnP-ADMM|26.55|0.865|24.93|0.812
> |Score-SDE|17.62|0.617|7.12|0.109
> |ADMM-TV |23.86|0.803|22.37|0.801
>
>
>
>
> | Method | SR(4X) |            | Gaussian Deblur |             |
> |:----------|:--------|:--------|:--------------------|:--------|
> |               |FID       |LPIPS |FID                        |LPIPS
> |PSLD (Ours)|34.28|0.201|41.53|0.221
> |DPS|39.35|0.214|44.05|0.257
> |DDRM|62.15|0.294|74.92|0.332
> |MCG|87.64|0.520|101.2|0.340
> |PnP-ADMM|66.52|0.353|90.42|0.441
> |Score-SDE|96.72|0.563|109.0|0.403
> |ADMM-TV |110.6|0.420|186.7|0.507
>
>
>
>
> ### Concluding Remark
>
> Please let us know if the clarifications and additions suitably address your concerns. We are happy to address any remaining points during the discussion phase.
>
>
>
> ### Reference
> [1] Song, Jiaming, Arash Vahdat, Morteza Mardani, and Jan Kautz. Pseudoinverse-guided diffusion models for inverse problems. In: International Conference on Learning Representations. 2023. url: https://openreview.net/forum?id=9_gsMA8MRKQ.
>
> [2] Yaniv Romano, Michael Elad, and Peyman Milanfar. The little engine that could: Regularization by
> denoising (RED). arXiv preprint arXiv:1611.02862, November 2016.
>
> [3] Saharia, C., Chan, W., Chang, H., Lee, C., Ho, J., Salimans, T., Fleet, D. and Norouzi, M., 2022, July. Palette: Image-to-image diffusion models. In ACM SIGGRAPH 2022 Conference Proceedings (pp. 1-10).
>
> [4] Bahjat Kawar, Gregory Vaksman, and Michael Elad. SNIPS: Solving noisy inverse problems
> stochastically. arXiv preprint arXiv:2105.14951, May 2021.
>
> [5] Yinhuai Wang, Jiwen Yu, and Jian Zhang. Zero-Shot Image Restoration Using Denoising Diffusion Null-Space Model. In: The Eleventh International Conference on Learning Representations. 2023. url: https://openreview.net/forum?id=mRieQgMtNTQ.
>
> [6] Meng, X. and Kabashima, Y., 2022. Diffusion model based posterior sampling for noisy linear inverse problems. arXiv preprint arXiv:2211.12343.

---

> > ### Comment · Reviewer_xKok · 2023-08-19
> > **Thanks for the rebuttal**
> >
> > I thank the authors for the rebuttal, especially the added comparisons with other methods, which further show the superiority of the proposed method. Please revise the future version accordingly. I have raised my score after reading the rebuttal.

---

### Official Review · Reviewer_S3JW · 2023-07-05

**Soundness:** 3 good
**Presentation:** 3 good
**Contribution:** 3 good
**Rating:** 5
**Confidence:** 4

**Summary:**

This paper focuses on solving inverse problems using diffusion based probabilistic models without **retraining**. The authors build on "Diffusion posterior sampling" which basically builds a diffusion model for the posterior using only the score of the prior distribution. Indeed, the score of the posterior $\nabla \log p_t(x_t | y)$ is equal to $\nabla \log p(y|x_t) + \nabla \log p_t(x_t)$ where the second term is given by the prior diffusion model. The first term is however intractable but can be approximated by noting that $p(y | x_t) = \int p(y|x_0) p(x_0 | x_t) dx_0$. This integral is then approximated by simply taking the mean a posteriori i.e. $p(y | x_t) \approx p(y | E(x_0 | x_t))$ which  itself can be approximated using Tweedie's formula and the prior score. The original DPS paper only considers the diffusion in the pixel space and this paper extends the methodology to latent diffusion models. While the direct extension is straightforward, the authors claim that it does not work in practice and they provide a rather intuitive modified posterior diffusion model that ensures consistency at the borders of the mask.



**Strengths:**

The paper tackles an important problem and provides a sound methodology for diffusion posterior sampling for latent space diffusion models. Furthermore, the numerical experiments suggest that this method outperforms other existing methods.

**Weaknesses:**

The fact that the authors show recovery in a 2 step diffusion model in a very basic setting proves nothing. First, the results hold only when the optimal solution is normalized. In realistic settings and with multi step diffusion one does not know in advance how to rescale (if even rescaling is the right to do?) the drift so that the diffusion model matches the target distribution. Next, the authors prove that DPS "provably" recovers the posterior but this to me is also meaningless since one can easily show numerically in a noisy inpainting setting with Gaussian mixtures (the posterior is tractable in this case) that DPS **does not** in fact sample the posterior. It samples outside the support of the posterior even in the simplest settings. I believe that the authors should provide a numerical experiment where the target posterior is known (again, Gaussian mixtures) and show that their method does sample from the posterior. To me there is no reason why it should exactly sample from the posterior and not outside the support like DPS. Indeed, this issue is not related to the quality of the generative model but more to the fact that DPS (and truthfully many of the difussion posterior sampling methods) are not exact samplers since there is no correction like in MCMC algorithms. I believe that the authors should drop the "provably" in the title as it is misleading.





**Questions:**

- What do the authors mean by the *curse of ambient dimension*, line 226-228? In DPS the gradient is indeed computed in the $d$-dimensional space but this is also the case of PSLD by the chain rule (and in fact the computational cost of PSLD is slightly larger).

- In Figure 2 right panel, an inpainting example is considered. In this case p(y|x_0) is a dirac delta so I am not sure how the authors manage to take the gradient. Do the authors approximate the dirac delta with a Gaussian with small variance? Was the same courtesy applied to DPS?

- It is quite surprising that in Algorithm 2 (but also Algo1) nothing depends on the measurement variance. Why is it not taken into account?

- While I could guess what it respresents, $x_0 *$ is not defined anywhere.

**Limitations:**

see below.

---

> ### Author Rebuttal · Authors · 2023-08-09
>
> ## Response to Reviewer S3JW
> Dear Reviewer S3JW,
>
> Thank you for the review and for pointing out the **importance of our work in leveraging the power of latent-based diffusion models** in solving inverse problems with state-of-the-art performance.
>
>
> Below, we provide answers to your remaining comments and questions.
>
>
> (1) **Recovery in two-step process is a basic setting.**
> The main purpose of the theoretical analysis is to give **intuition** (as rightly pointed out by Reviewer vbWT) on why gluing term is critical to the empirical success of PSLD. The two-step process in a linear model setting **serves this purpose without unnecessary mathematical complications**. To elaborate, (a) vanilla extension of DPS fails due to many-to-one mapping of the VAE encoder, (b) GML-DPS fails due to infinitely many fixed-points of the linear system, and (c) PSLD works due to its inherent nature to find the stable fixed-point. The theoretical analysis gave us the intuition for the gluing objective that led to contraction of the distance from optimal solution and thereby strong empirical performance.  Besides, the main contribution of the paper is to solve inverse problems using latent diffusion models, unlocking the potential of large foundation models (e.g. Stable Diffusion) as rightly pointed out by Reviewer  vbWT, yVXi and xKok. We substantiate our contribution with large-scale experiments typically considered in practice.
>
> (2) **One does not know in advance how to scale.**
> We would like to point out that the scaling factor is a hyper-parameter that is usually tuned in practice. Our theory provides intuitions on how to better tune this factor and rule out some unwanted experiments. Besides, we believe this is an issue of the generative model, not posterior sampling. As long as we have a pre-trained generative model that can sample from the data distribution, our method can leverage this model for posterior sampling.
>
>
> (3) **DPS does not sample the posterior in noisy setting.** The setting we consider is noiseless and exactly recoverable, where we prove that DPS samples the posterior under valid assumptions. We will clarify this noiseless setting in the revised version.
>
>
> (4) **Many diffusion posterior sampling algorithms are not exact samplers.**
> Many diffusion posterior samplers are not exact samplers, but their empirical performance is significantly better than any of the MCMC algorithms with strong theoretical guarantees. To the best of our knowledge, these MCMC algorithms with strong theoretical results are only shown to work in toy experimental setting. **Provably in title.** To remove ambiguity, we will change the title to *"Solving **Linear** Inverse Problems Provably via Posterior Sampling with Latent Diffusion Models"*.
>
> (5) **What do the authors mean by the curse of ambient dimension?**
> As we discuss in Section 3, PSLD computes the gradients in the latent space with dimension $k$ (wrt $\mathbf{z}_i$), whereas DPS computes the gradients in the pixel space with dimension $d$ (wrt $\mathbf{x}_i$). In practice, the computational complexity of Stable Diffusion model ($\sim 4.00GB$) is higher (roughly 6 times) than the computational complexity of the encoder-decoder model ($\sim 700 MB$). Therefore, applying the chain rule in the encoder-decoder and running diffusion in the latent space is less expensive than applying diffusion models in the pixel space directly.
>
> (6) **How to take gradients of $P(y|x_0)$ in block inpainting?**
> As correctly pointed out by the reviewer, we use Gaussian approximation of the dirac delta. The same idea is applied in DPS approximation for handling block inpainting.
>
> (7) **It is quite surprising that in Algorithm 2 (but also Algo1) nothing depends on the measurement variance. Why is it not taken into account?**
> We believe there is a misunderstanding. Recall that this is a maximum likelihood estimation problem. Since $\mathbf{y} = AD(\mathbf{z})+\sigma_y\mathbf{n} $, where $\mathbf{n}\sim \mathcal{N}(\mathbf{0},\mathbf{I})$, we have $\mathbf{y} \sim P(y|z) \propto \exp{\left(-\frac{1}{2\sigma_y^2} || \mathbf{y} - AD(\mathbf{z})||_2^2\right)}$. Taking the gradient of the log-likelihood, we have $\eta\nabla_z || \mathbf{y} - AD(\mathbf{z})||_2^2$, where $\eta$ absorbs the scaling factor $\frac{1}{2\sigma_y^2}$.
>
>
> (8) **Definition of $x_0^\*$.**
> Thank you for pointing out the typo. We will correct it in the revision. We denote by $x_0^*$ the true underlying sample as you could have rightly guessed.
>
>
>
> ### Concluding Remark
>
> Please let us know if the clarifications and additions suitably address your concerns. We are happy to address any remaining points during the discussion phase.

---

> > ### Comment · Reviewer_S3JW · 2023-08-15
> >
> > I would like to thank your for your response. I still have some disagreements on certain points:
> >
> > **DPS does not sample the posterior in noisy setting -  Many diffusion posterior sampling algorithms are not exact samplers.** What I meant is that the result proven for DPS in the paper does not have any practical implication; as soon as one departs from the assumptions made in the paper, DPS fails to sample from the simplest posterior. The authors can perhaps try it on simple toy examples; take a $d$-dimensional Gaussian mixture and observe only one coordinate for example. DPS will sample inside but also outside of the support of the posterior and there is no evidence on why this shouldn't be the case of the proposed algorithm. Hence, the assumptions made in the paper are very strong and yield results with questionnable meaning.
> >
> > Regarding the comment on MCMC, I strongly disagree with the conclusion. Assuming that MCMC algorithms only work on toy algorithms, methods such as DPS do not work on such toy examples and as such, there is no reason to believe that they sample approximately from the correct posterior in high dimensional examples if they fail on the simplest examples. The quantitative metrics that are usually used like the FID and LPIPS mean nothing in practice; they do not assert the quality of posterior sampling. They quantify how coherent the images are but not if they're accurate samples from the posterior. It is practically impossible to know if methods such as DPS or PSLD sample from the posterior as we do not have access to samples from the said posterior. The only way to get a rough idea is to use these methods on examples for which the posterior is available in closed form.
> >
> > I have one remaining question; from my experience DPS is very sensitive to the "step-size" $\zeta_i$. It can yield very bad results when chosen inappropriately. In the main paper there does not seem to be any discussion on this matter for your methods and I believe that this is quite important. Did the authors tune the parameter $\eta$? Furthermore, I understand that in the derivations the noise std is factored into $\eta$. But what  I meant is that intuitively $\eta$ should depend, in the experiments, on the noise std. It seems to me that the noise std is the same on all experiments and that this matter is not discussed.

---

> > > ### Author Response · Authors · 2023-08-16
> > > **Discussion with Reviewer S3JW**
> > >
> > > Dear Reviewer S3JW,
> > >
> > >
> > > (1) **DPS does not sample the posterior in noisy setting.**
> > > Thanks for the comment, we now better understand your intent. We agree with your comment on DPS, and our result only holds in the linear noiseless case under restrictive assumptions. We will emphasize this in the paper.  However, this stylized theory did produce one practically useful result: We wanted to solve linear inverse problems in the latent space to leverage the power of Stable diffusion and other pre-trained foundation models. A standard way of using DPS in the latent space failed, and this led us to design the gluing objective. Specifically, even in the linear noiseless case, (latent) DPS did not recover the ground-truth (due to lack of contraction to the unique fixed point), motivating the gluing objective to fix this issue.
> > > As we responded to reviewer yVXi, we will remove the pixel-space analysis (move it to the appendix), shorten the latent-space analysis, and emphasize it only holds for the noiseless case. We will further highlight that the analysis goal is to provide intuition on the gluing objective.
> > >
> > > (2) **Sampling from true posterior and metrics such as LPIPS and FID.**
> > > This is a fair point. There is no theoretical justification to assert that DPS or PSLD sample from the true posterior. However, in our paper we study the solution of inverse problems where the ground truth is known. This helps in validating the reconstruction produced by our algorithm, and thus pairwise error-metrics like MSE/PSNR and LPIPS can be used. These metrics are used, e.g. for MRI quality assessment and numerous other inverse problems, as non-perfect but still reasonable metrics to compare inverse problem solvers. Finally, as you rightly point out, there is no theory showing that metrics like FID or Inception score say anything about the quality of posterior sampling and this remains as an important research direction.
> > >
> > >
> > > (3) **Did the authors tune the parameter $\eta$? In the main paper there does not seem to be any discussion on this matter for your methods and I believe that this is quite important.** We tuned the hyper-parameter $\eta$ to the extent possible with our available computing resources. The reported results are with the best hyper-parameter we could find. We will revise the discussion to make this clear.
> > >
> > >
> > > (4) **It seems to me that the noise standard deviation is the same on all experiments and this matter is not discussed.** As rightly pointed out by the reviewer, the noise standard deviation is the same on all experiments. This is the same setting as the baseline DPS. We will add this discussion in the revised version.
> > >
> > >
> > >
> > > ### Concluding Remark
> > > We are happy to address any remaining points during the discussion phase.

---

### Official Review · Reviewer_yVXi · 2023-07-06

**Soundness:** 3 good
**Presentation:** 3 good
**Contribution:** 3 good
**Rating:** 7
**Confidence:** 3

**Summary:**

While the preivous methods have focused on solving linear inverse problems based on diffusion models, the paper presents a first extension to latent diffusion models (LDM). The core idea is developed upon the existing DPS method, which forms an approximation to p(y|xt) by using the denoising score estimate. The key challenges of the extension to LDM are (1) the non-bijective mapping between encoder and decoder, and (2) the inconsistency at the boundary of mask (for inpainting problems). To address these challenges, the authors propose an additional gluing term that penalizes the discontinuity at the mask boundary.

The main practical contribution of this paper is that if unlocks the potential of large pre-trained LDM (e.g. stable diffusion), and hence the proposed based based on more powerful pre-trained models enjoys superior performance compared to its counterpart with ambient / pixel space settings.

In addition, the paper also presents a theoretical analysis in a toy setting (linear two-step diffusion models) to motivate the proposed objective.

**Strengths:**

- While the extension of DPS to LDMs seems natural, I appreciate the practical impact of the proposed methodology given the popularity and superior power of existing pre-trained LDMs.

- If I understand it correctly, doing conditional inference in latent space would also be more efficient than doing inference in the ambient space (after controlling the model complexity though in practice I wonder if that comparison can be made)

- The proposed method is also claimed to be robustness of the choice of step-size, at least in toy settings, though I am not sure how this would extend to general case.



**Weaknesses:**

- The theoretical analysis seems to follow reference 30 closely, takes up a large portion of the paper. However, I am not sure how much insights can be carried over to the general cases. In my opinion, the proposed method stays well-motivated without this analysis. Hence. I believe either discussing the extension to general cases, or trimming this part would make a better paper.

- The evaluation part can be misleading given that pre-trained LDMs are "trained on much more data compared to the one used by DPS" (ref: appendix, caption above table 5). I think this information should be disclosed in the main paper instead of the appendix. In addition, it would be helpful to also disclose the unconditional generation performance of all backbone diffusion models, as well as the size of the models, the number of training data, etc.


**Questions:**

- Theorem 3.8: what is the value for gamma_i in Alg 2?

- Theorem 3.8 and its surrounding text: can you elaborate more on why it is robust to different values of step size eta? In the equation between line 221 and line 222 on page 7, apparently if I choose different value of eta, the resulting z0 would be different (as long as the grad norm after eta is non-zero). Am I missing anything here? In addition, in this equation, should there be a multiplicative factor \gamma before the last term, or is it set to 1 for the theorem and the statement to be correct?

- Page 7. line 228 "DPS algorithm suffers from the curse of ambient dimension". Can you elaborate on the issue? Is it mainly the efficiency issue or the approximation accuracy issue?

- Would be helpful to accomplish the theoretical analysis with toy data empirical verification.

- (If possible), would be helpful to provide results where the base diffusion models for DPS and PSLD have similar performance.

**Limitations:**

Yes.

---

> ### Author Rebuttal · Authors · 2023-08-09
>
> ## Response to Reviewer yVXi
> Dear Reviewer yVXi,
>
> Thank you for the review and for pointing out the fact that our study proposes the **first method** for inverse problems with latent diffusion models, and **unlocks the potential** of large **pre-trained LDMs** for sample recovery.
>
> Below, we respond to your remaining comments and questions.
>
> (1) **Methodology stays well-motivated, trimming theoretical analysis would make a better paper.**
> We will trim this section of the paper as per the reviewer's comment. Note that the main purpose of the theoretical analysis is to give **intuition** (as rightly pointed out by Reviewer vbWT) on why gluing term is critical to the empirical success of PSLD. The two-step process in a linear model setting **serves this purpose without unnecessary mathematical complications**. To elaborate, (a) vanilla extension of DPS fails due to many-to-one mapping of the VAE encoder, (b) GML-DPS fails due to infinitely many fixed-points of the linear system, and (c) PSLD works due to its inherent nature to find the stable fixed-point. The theoretical analysis gave us the intuition for the gluing objective that led to contraction of the distance from optimal solution and thereby strong empirical performance.
>
> (2) **"Stable Diffusion is trained on much more data compared to the DM used by DPS". Move this information from Table 5 in appendix to the main body of the paper.**
> As suggested, we will move this statement to the main body of the paper. However, we would like to highlight that the focus of the paper is to **unlock** the potential of large pre-trained LDMs (e.g. Stable Diffusion), not to beat DPS.
>
> (3) **Helpful to discuss unconditional generation performance of the backbone diffusion models, dataset, size etc.**
> Since most of the large pre-trained LDMs (e.g. Stable Diffusion) are maintained by commercial service providers, they keep updating their pre-trained weights, datasets, and training iterations.  It is hard to compare these generative models with the pixel-space diffusion model used by DPS. Nevertheless, we report the metrics on FFHQ from the original papers and the github source code.
>
>
> | Model    | Dataset|Weights|Iterations |Image Size|
> |:---------|:-------|:------|:----------|:---------|
> |SD-v-1.5  |LAION-5B|4.00GB |840K       | 512x512  |
> |PSLD-LDM  |FFHQ    |2.40GB |635K       | 256x256  |
> |DPS-DM    |FFHQ    |358MB  |1M         | 256x256  |
>
>
>
> The FID score of an earlier version of LDM is 4.98 [1]. We could not find the unconditional generation performance of DPS in terms of FID anywhere as the authors did not evaluate it for generative modeling. The goal of their paper was to study **posterior sampling** using a pre-trained diffusion model. Also, sampling from diffusion models takes much more time than sampling from other generative models, such as GANs, which could be another reason for lack of evaluation in generative modeling unless that is the main contribution.
>
>
>
> (5) **Theorem 3.8: What is the value of $\gamma_i$ in Alg 2?**
> We set $\gamma_i=1$ as it is immaterial in a linear model setting.
>
> (4) **Intuition behind Theorem 3.8 and its surrounding text.**
> The intuition is that, after every step of denoising and measurement update, we would like to solve an optimization problem to make sure that the decoded sample resides on the manifold of natural images. This is achieved by the gluing update. In practice, one step of gluing update suffices instead of exactly solving the optimization problem at every step. However, in a perfectly recoverable linear model setting, the optimization problem can be exactly solved in one step. This makes the gluing term robust against the choice of step size $\eta$.
>
> As we discuss in line 184, the multiplicative factor $\gamma$ is immaterial in a linear model setting. Hence, we set it to $\gamma=1$. Empirically, we observe that $\gamma=0.1$ and $\eta=1$ are robust to several downstream tasks given in Appendix B (please also see newly added experiments in the attached PDF).
>
>
> (6) **Elaborate on "DPS suffers from the curse of ambient dimension".**
> DPS suffers from curse of ambient dimension because in this method, gradients are computed in the pixel space with dimension $d$. However, latent-based methods such as PSLD compute gradients in the latent dimension $k$, and hence the computation is more efficient. Furthermore, applying the chain rule on VAE and running diffusion in the latent space is less expensive than running diffusion in pixel space directly.
>
> (7) **(If possible) Compare PSLD and DPS with the same diffusion model.**
> Since our goal is to solve inverse problems using LDMs, we use large pre-trained LDMs (e.g., Stable Diffusion) as our generative prior. In practice, training these LDMs require a lot of computational investment and their performance is usually better than pixel-space diffusion models. Although we would be happy to compare their performance with the same diffusion model, we are not aware of any pixel-space diffusion model that is comparable to latent-space diffusion models.
>
>
> ### Concluding Remark
>
> Please let us know if the clarifications and additions suitably address your concerns. We are happy to address any remaining points during the discussion phase.
>
> ### Reference
> [1] Rombach, Robin, Andreas Blattmann, Dominik Lorenz, Patrick Esser, and Björn Ommer. "High-resolution image synthesis with latent diffusion models." In Proceedings of the IEEE/CVF conference on computer vision and pattern recognition, pp. 10684-10695. 2022.

---

> > ### Comment · Reviewer_yVXi · 2023-08-17
> >
> > Thank the authors for their thorough and thoughtful response. All my concerns have been touched and addressed.

---

### Official Review · Reviewer_vbWT · 2023-07-06

**Soundness:** 3 good
**Presentation:** 2 fair
**Contribution:** 3 good
**Rating:** 7
**Confidence:** 4

**Summary:**

This paper investigates the use of diffusion models for solving inverse problems. While many recent papers have explored diffusion models for inverse problems, to the best of my knowledge, this study is the first to propose a method for inverse problems with latent diffusion. One advantage of latent diffusion is its lower computational demand.

The main challenge in solving inverse problems with diffusion models lies in computing the intractable likelihood term $\nabla \log p_{x_t}( y | x_t )$, where $x_t$ represents a noisy point in pixel-space. Recently, the DPS algorithm was introduced to approximate this term as $\nabla \log p_{x_t}( y | \mathbb{E}[x_0| x_t] )$. The first contribution of this paper is to extend the DPS framework to a latent diffusion model, approximating the intractable likelihood $\nabla \log p_{x_t}( y | z_t )$, where $z_t$ denotes a noisy point in latent space, with $\nabla \log p_{z_t}( y | \mathbb{E}[z_0| z_t] )$ along with an additional term that enforces $\mathbb{E}[z_0| z_t]$ to be a fixed point of the autoencoder (GML-DPS). Furthermore, the latter term is modified to enforce consistency in the measurements (PSLD). The empirical performance of the PSLD algorithm is compared with the standard DPS in high-dimensional reconstruction tasks, and improvements over DPS are demonstrated.

These three algorithms (DPS, GML-DPS, PSLD) are theoretically studied in the case of a data distribution corresponding to a Gaussian supported on a low-dimensional subspace, perfect knowledge of the subspace, zero noise in the measurements, and a measurement matrix that is bijective over the subspace. Specifically, this implies that the inverse problem can be exactly solved given the measurements. Finally, the sampling processes are two-step diffusion processes. The authors demonstrate that all three algorithms precisely recover the target signal, and PSLD exhibits robustness to variations in the specification of the step sizes.

**Strengths:**

Solving inverse problems with (large) latent diffusion models is a relatively underexplored area, and this paper fills a gap in the existing literature. In particular, the authors demonstrate how their methods can be used with SOTA latent-based foundation models such as stable diffusion. The empirical results presented are promising and demonstrate the potential of this approach.

Additionally, the authors offer some theoretical guarantees, albeit in a limited setting. These guarantees are valuable and noteworthy, considering the field's scarcity of rigorous results, even in simple toy examples.

**Weaknesses:**


The main weaknesses of the paper lie in the experimental evaluations and results. Specifically, all experiments are conducted on subsets of the FFHQ dataset and images sourced from the internet. The authors evaluate the PSLD algorithm on two latent models: LDM-VQ-4, trained on FFHQ, and Stable Diffusion, trained on a significantly larger dataset (LAION). However, the comparisons are made between PSLD on these two latent models  (proposed) and DPS (baseline) on a standard diffusion model trained *exclusively* on FFHQ.

The authors demonstrate improved performance of PSLD on the (latent) Stable Diffusion model compared to DPS on the standard diffusion model. However, when employing the latent model LDM-VQ-4 trained solely on FFHQ, the improvements are marginal (Table 3).

As a result, it becomes challenging to ascertain whether the observed improvements stem from algorithmic enhancements or simply the utilization of superior models trained on larger datasets, which may also account for the enhanced out-of-distribution performance.

Furthermore, the experiments do not shed light on the improvement attributable to the addition of the "gluing" term in comparison to the DPS's vanilla extension term. One might hypothesize that the "gluing" term could be employed in DPS (by removing the encoder-decoder), potentially enhancing the algorithm's performance.



**Questions:**

Some typos and questions:

- In Section 2, the notation can be confusing. To clarify, it is suggested to use $x_0^*$ to denote the target signal and distinguish it from the first sample of the backward diffusion ($x_0$).

- For Equations (3), should it be $\hat{x}_0 = \mathbb{E}[x_0 | x_t]$?

- In Algorithm 1: line 1 what is $\mathcal{T}$?

- Equations (7) should include a $\log$ term for the DPS vanilla extension. Additionally, as shown in the supplementary material, $\mathcal{A}^\top$ should be added to (7) and line 8 of Algorithm 2.

- In Algorithm 2, it appears that the unknown target $x_0^*$ is needed, but in reality, the algorithm only requires the measurements $y$. It is advisable to modify the algorithm to reflect this (as in Algorithm 1).

- Line 142 of the Theoretical section introduces the noisy inverse problem as $y = A x_0 + \sigma_y n$.  However, it should be noted that exact recovery is not possible in this case and it seems that the theoretical results are proven under the assumption of $\sigma_y = 0$, where the problem can be solved exactly. The authors should explicitly address these points to ensure clarity.

**Limitations:**

Yes

---

> ### Author Rebuttal · Authors · 2023-08-09
>
> ## Response to Reviewer vbWT
>
> Dear Reviewer vbWT,
>
>
> Thank you for highlighting the fact that our study proposes the **first method** for inverse problems with latent diffusion models. We also thank you for your comment that the **theoretical results are valuable and noteworthy** and the **empirical results are promising**.
>
> Below, we respond to your remaining comments and questions.
>
> (1) **The main weakness is that all experiments are conducted on subsets of FFHQ and images from the internet.** To make a fair comparison with the baselines, we use the same experimental setting and evaluate on the same subsets of FFHQ as the baselines. In addition, we show improved performance on out-of-distribution images from the internet. We view this as a strength because we use the same generative model for both the datasets, whereas prior works (including DPS) require dataset specific generative models.
>
> (2) **We only show marginal improvement of PSLD (LDM-VQ-4) over DPS in Table 3.**
> Our goal is to develop a framework to leverage the power of pretrained latent models, such as Stable Diffusion. In this table, our focus is not to maximize the margin that we beat the previous state-of-the-art DPS. The fact that we can still obtain results comparable to (or better than) the state-of-the-art DPS using an LDM (see Table 3) indicates that we do not loose a lot of information by shifting the diffusion process to the lower-dimensional latent space. Latent diffusions are much faster, and also our method
> unlocks the potential of using pre-trained LDMs as pointed out by Reviewer yVXi. Besides, Table 2 (also the newly added results) shows that PSLD can be better than DPS by simply scaling these LDMs, which is a common practice adopted by commercial service providers.
>
> (3) **Is our Improvement due to better algorithm or better generative models due to larger training dataset?**
> We would like to clarify that the significant gain of PSLD over DPS (both in-distribution and out-of-distribution) is partly due to the fact that Stable Diffusion has been trained on a significantly larger dataset. However, it is important to note that none of the existing posterior sampling algorithms could leverage this pre-trained latent diffusion model for inverse problems. Another important aspect is the ability of PSLD to use the same generative model for several downstream tasks on FFHQ, ImageNet, and random images sourced from the web. On the contrary, prior works (including DPS) require  dataset specific generative models, which limits their application to general domain images.
>
> (4) **Is our improvement attributable to gluing term, compared to the DPS vanilla extension?**
> As we discuss in Section 2.1, the vanilla extension of DPS fails due to many-to-one mapping of the encoder. Still, we believe this is not a fair comparison with DPS because the authors originally built DPS for a pixel-space diffusion model (not a latent one). Making DPS approximation work in the latent-space requires non-trivial extensions, which we discuss in Section 2.1, Section 3.2 (**Theorem 3.4**), and Section 3.3 (**Theorem 3.7 and 3.8**).  We also provide experimental results in Appendix B.2 (Table 5) and newly added results on super-resolution and Gaussian deblur tasks (please see the attached pdf).
>
> (5) **One may hypothesize that including gluing term in DPS may enhance its performance.**
> Naively gluing in pixel-space will create visible edges separating inpainted pixels from the observed ones. Still, the gluing term in pixel-space DPS will provide guidance through an extra gradient term $\nabla_{\mathbf{x}_i}||A^T\mathbf{y} - \hat{\mathbf{x}}_0||_2^2$. This is an interesting direction for future research, but not within the scope of our paper since our goal is to solve inverse problems using latent diffusion models.
>
> (6) **Typos and questions.** We will correct all the typos in the revised version, thank you for your careful reading.
> For equation (3), it should be $\hat{x}_0 = \mathbb{E}[x_0|x_t]$. In Algorithm 1: $\mathcal{T}$ in line 1 should be the standard normal distribution $\mathcal{N}$. Since our algorithm only requires $\mathbf{y}$, we will modify the **Input** in Algorithm 2 similar to DPS as suggested by the reviewer. **(Line 142)** As correctly pointed out by the reviewer, exact recovery is possible when $\sigma_y=0$ (the setting we consider), but not when $\sigma_y>0$. We will clarify this in the revision.
>
>
> ### Concluding Remark
>
> Please let us know if the clarifications and additions suitably address your concerns. We are happy to address any remaining points during the discussion phase.

---

> > ### Comment · Reviewer_vbWT · 2023-08-14
> >
> > I appreciate the thorough response provided by the authors. I believe that this paper constitutes a valuable contribution and increased my score.
> >
> > However, I would like to discuss the phrasing and structure of the experiments in Section 5. I find it slightly misleading, which not only led me astray but possibly other reviewers as well. I understand the enthusiasm surrounding the outcomes achieved with the stable diffusion model. However, I believe it would be more equitable for the readers if the authors first presented and discussed the results for PSLD on the latent diffusion model LDM-V Q-4. Comparing these outcomes with those of DPS for a standard diffusion model makes sense, as these two models are trained using the same volume of data. This approach would effectively establish the authors' assertion that the primary innovation lies in the proposed algorithm for latent diffusion, which can perform comparably well, if not slightly better, than the state-of-the-art DPS for standard diffusion models. Following these initial experiments, presenting the results for the Stable diffusion model would be logical and would further reinforce the authors' argument.
> >
> > Finally, I am assuming that the authors have tested equation (5), subsequently moving on to (6), before eventually arriving at the proposed (7). I am curious whether the authors have conducted experiments to demonstrate that using (6) indeed outperforms the basic (5), and also to showcase that adopting (7) results in improvements over (6). It is possible that this information is present in the appendices, but I may have overlooked it.

---

> > > ### Author Response · Authors · 2023-08-15
> > > **Discussion with Reviewer vbWT**
> > >
> > > Dear Reviewer vbWT,
> > >
> > > Thank you for reading our response and increasing the score. We greatly appreciate your timely feedback and active participation during the discussion phase. Below, we respond to your new comments.
> > >
> > > (1) **Discussion on the phrasing and structure of the experiments in Section 5.** We thank the reviewer for the suggestion on how to make the flow of the experimental Section 5 more logical and further reinforce our arguments. As suggested by the reviewer, we will rephrase and restructure Section 5 in the revised version of the paper.
> > >
> > > (2) **Have the authors tested equation (5), (6) and (7)?** We have tested equation (5), (6) and (7) leading up to the main idea of PSLD. In our inpainting experiment, we found that equation (5) failed to generate coherent images at the boundary, equation (6) made the boundary smooth at the cost of extensive parameter tuning, and finally equation (7) mitigated these issues that resulted in high PSNR and SSIM. Quantitatively, we have already demonstrated the improvement of equation (7) over equation (6) in the Appendix B.2 (Table 5). Also, the newly added experiments in the attached PDF (Table 1) shows this improvement of equation (7) over equation (6). In the revised version, we will add experiments with equation (5) as another baseline.
> > >
> > >
> > > ### Concluding Remark
> > > We are happy to address any remaining points during the discussion phase.

---

### Author Rebuttal · Authors · 2023-08-09

### Response to all reviewers
Dear Reviewers,

We thank you for carefully reading our paper and providing us with valuable feedback. Below, we summarize the reviews and newly added experiments to substantiate our contributions.

(1) We are encouraged by the **unanimous comment** by all reviewers that our study proposes the **first method for solving inverse problems with LDMs**, which **unlocks** the potential of large pre-trained LDMs, such as Stable Diffusion.

(2) We thank Reviewer vbWT for highlighting the fact that the **theoretical result is valuable and noteworthy**, considering the field's scarcity of rigorous results, even in toy examples. We also thank all the reviewers for pointing out the fact that our **experimental results are promising**.

(3) Regarding constructive feedback, we have **favorably addressed all the questions** raised by the reviewers. In particular, we thank Reviewer vbWT, yVXi and xKok for suggesting relevant experiments with quantifiable metrics, which helped **strengthen our contributions**. We have added the following experimental results in the attached PDF:

    (a) Quantitative results on Super-resolution (4X): Table 1
    (b) Quantitative results on Gaussian Deblur: Table 1
    (c) Qualitative results on colorization: Figure 1
    (d) Quantitative comparison of runtimes of different algorithms: Table 2
    (e) Quantitative comparison of NFEs of different algorithms: Table 2
    (f) Quantitative comparison of unconditional generation performance: Table 2
    (g) Overall pipeline of our proposed framework for arbitrary mask: Figure 2

(4) We thank Reviewer xKok for bringing related works to our attention. We have compared with these methods and cited accordingly in the revised version.



### Concluding Remark

Please let us know if the clarifications and additions suitably address your concerns. We are happy to address any remaining points during the discussion phase.



### Reference

[1] Song, Jiaming, Arash Vahdat, Morteza Mardani, and Jan Kautz. Pseudoinverse-guided diffusion models for inverse problems. In: International Conference on Learning Representations. 2023. url: https://openreview.net/forum?id=9_gsMA8MRKQ.

[2] Yaniv Romano, Michael Elad, and Peyman Milanfar. The little engine that could: Regularization by
denoising (RED). arXiv preprint arXiv:1611.02862, November 2016.

[3] Saharia, C., Chan, W., Chang, H., Lee, C., Ho, J., Salimans, T., Fleet, D. and Norouzi, M., 2022, July. Palette: Image-to-image diffusion models. In ACM SIGGRAPH 2022 Conference Proceedings (pp. 1-10).

[4] Bahjat Kawar, Gregory Vaksman, and Michael Elad. SNIPS: Solving noisy inverse problems
stochastically. arXiv preprint arXiv:2105.14951, May 2021.

[5] Yinhuai Wang, Jiwen Yu, and Jian Zhang. Zero-Shot Image Restoration Using Denoising Diffusion Null-Space Model. In: The Eleventh International Conference on Learning Representations. 2023. url: https://openreview.net/forum?id=mRieQgMtNTQ.

[6] Meng, X. and Kabashima, Y., 2022. Diffusion model based posterior sampling for noisy linear inverse problems. arXiv preprint arXiv:2211.12343.

---

### Decision · Program_Chairs · 2023-09-21

**Decision:**

Accept (poster)

**Comment:**

This paper proposes a new method to deal with linear inverse problems using pre-trained latent diffusion models (LDM). It lowers computational demand for large-scale problems, and unlocks the potential of large pre-trained LDM for inverse problem solving.
Most of the reviewers acknowledge the novelty of the method, along with the theoretical justifications.

Please incorporate all the reviewers' feedback into the final version, especially the new experimental comparison with other methods.